# A single Ho-induced double-strand break at the *MAT* locus is lethal in *Candida glabrata*

**Laetitia Maroc**[1], **Youfang Zhou-Li**[1], **Stéphanie Boisnard**[2], **Cécile Fairhead**[1]*

**1** Université Paris-Saclay, INRAE, CNRS, AgroParisTech, GQE—Le Moulon, Gif-sur-Yvette, France,
**2** Université Paris-Saclay, CEA, CNRS, Institute for Integrative Biology of the Cell (I2BC), Gif-sur-Yvette, France

* cecile.fairhead@universite-paris-saclay.fr

**Data Availability Statement:** All relevant data are within the manuscript and its Supporting Information files.

**Funding:** LM is recipient of a PhD fellowship from the French "Ministère de l'Enseignement Supérieur,

## Abstract

Mating-type switching is a complex mechanism that promotes sexual reproduction in Saccharomycotina. In the model species *Saccharomyces cerevisiae*, mating-type switching is initiated by the Ho endonuclease that performs a site-specific double-strand break (DSB) at *MAT*, repaired by homologous recombination (HR) using one of the two silent mating-type loci, *HMLalpha* and *HMRa*. The reasons why all the elements of the mating-type switching system have been conserved in some Saccharomycotina, that do not show a sexual cycle nor mating-type switching, remain unknown. To gain insight on this phenomenon, we used the yeast *Candida glabrata*, phylogenetically close to *S. cerevisiae*, and for which no spontaneous and efficient mating-type switching has been observed. We have previously shown that expression of *S. cerevisiae*'s Ho (*Sc*Ho) gene triggers mating-type switching in *C. glabrata*, but this leads to massive cell death. In addition, we unexpectedly found, that not only *MAT* but also *HML* was cut in this species, suggesting the formation of multiple chromosomal DSBs upon *HO* induction. We now report that *HMR* is also cut by *Sc*Ho in wild-type strains of *C. glabrata*. To understand the link between mating-type switching and cell death in *C. glabrata*, we constructed strains mutated precisely at the Ho recognition sites. We find that even when *HML* and *HMR* are protected from the Ho-cut, introducing a DSB at *MAT* is sufficient to induce cell death, whereas one DSB at *HML* or *HMR* is not. We demonstrate that mating-type switching in *C. glabrata* can be triggered using CRISPR-Cas9, without high lethality. We also show that switching is Rad51-dependent, as in *S. cerevisiae*, but that donor preference is not conserved in *C. glabrata*. Altogether, these results suggest that a DSB at *MAT* can be repaired by HR in *C. glabrata*, but that repair is prevented by *Sc*Ho.

## Author summary

Mating-type switching is one of the strategies developed by fungi to promote sexual reproduction and propagation. This mechanism enables one haploid cell to give rise to a cell of the opposite mating-type so that they can mate. It has been extensively studied in the yeast *S. cerevisiae* in which it relies on a programmed double-strand break performed by the Ho endonuclease at the *MAT* locus which determines sexual identity. Little is

de la Recherche et de l'Innovation". The funders had no role in study design, data collection and analysis, decision to publish, or preparation of the manuscript.

**Competing interests:** The authors have declared that no competing interests exist.

known about why the mating-type switching components have been conserved in species like *C. glabrata*, in which neither sexual reproduction nor mating-type switching is observed. We have previously shown that mating-type switching can be triggered, in *C. glabrata*, by expression of the *HO* gene from *S. cerevisiae* but this leads to massive cell death. In this work, we show that mating-type switching in *C. glabrata* can be triggered by CRISPR-Cas9 and without any high lethality. We demonstrate that the cut at *MAT* is only lethal when the Ho endonuclease performs the break, a situation unique to *C. glabrata*. Our work points to a degeneration of the mating-type switching system in *C. glabrata*. Further studies of this phenomenon should shed light on the evolution of mating systems in asexual yeasts.

## Introduction

In eukaryotes, sexual reproduction is a nearly ubiquitous feature and implies fundamental conserved processes such as gamete fusion, zygote formation and meiosis [1]. Sexual reproduction leads to genetic recombination between organisms and thus enables them to purge their genomes from deleterious mutations, as well as to increase their genetic diversity. It is in the fungal kingdom that the greatest diversity of sexual reproduction is found [1]. Particularly, sexual reproduction in fungal pathogens of human exhibits a considerable plasticity between species [2,3]. While many were thought to be asexual, several atypical sexual or parasexual cycles have been discovered. It has been shown that the yeast *Candida albicans* can perform a parasexual cycle by mating of two diploid cells, forming a tetraploid, that can undergo chromosome loss [4]. The more distant filamentous opportunistic pathogen, *Aspergillus fumigatus* exhibits a sexual cycle but only mates after spending 6–12 months in the dark [5]. Altogether, this suggests that, in most fungi, performing genetic exchange is crucial, even in well-adapted human pathogens.

In fungi, sexual reproduction can occur through three mechanisms [1]: heterothallism (requiring two compatible partners for mating to occur), homothallism (self-fertility), and pseudo-homothallism (where a single individual can go through a complete sexual cycle but mating only occurs between two compatible partners). Pseudo-homothallism has mainly been described in ascomycete yeasts where it occurs through a programmed differentiation process called mating-type switching [6]. This mechanism enables one haploid cell to give rise to a cell of the opposite mating-type so that they can mate. In all cases studied so far, it implies a genomic DNA rearrangement of the mating-type locus (*MAT*, encoding the key regulators of sexual identity) and species have evolved very different molecular pathways for the same aim. In the fission yeast *Schizosaccharomyces pombe*, an imprint at *mat1* (it is unknown whether the imprint is an epigenetic mark or a single nick) is introduced, that leads to a DSB during DNA replication [7,8]. Repair occurs with one of the two silent copies of *mat1*, called *mat2* and *mat3*. In the ascomycete *Kluyveromyces lactis*, mating-type switching involves a DSB at *MAT* but it is performed by two specific nucleases depending on the mating-type of the cell [9,10]. Mating-type switching has been extensively studied in the model yeast *S. cerevisiae* and has notably allowed a better understanding of cell identity, DSB repair and silencing mechanisms [11].

In *S. cerevisiae*, haploid cells can be of either mating-type, *MATalpha* or *MATa*, which encodes "alpha" or "a" information, respectively, at the Y sequence of the *MAT* locus [12] (Fig 1). Mating-type switching relies on a programmed DSB at the *MAT* locus performed by the Ho endonuclease at its 24-bp recognition site (Fig 1). DSBs are highly toxic DNA lesions, and

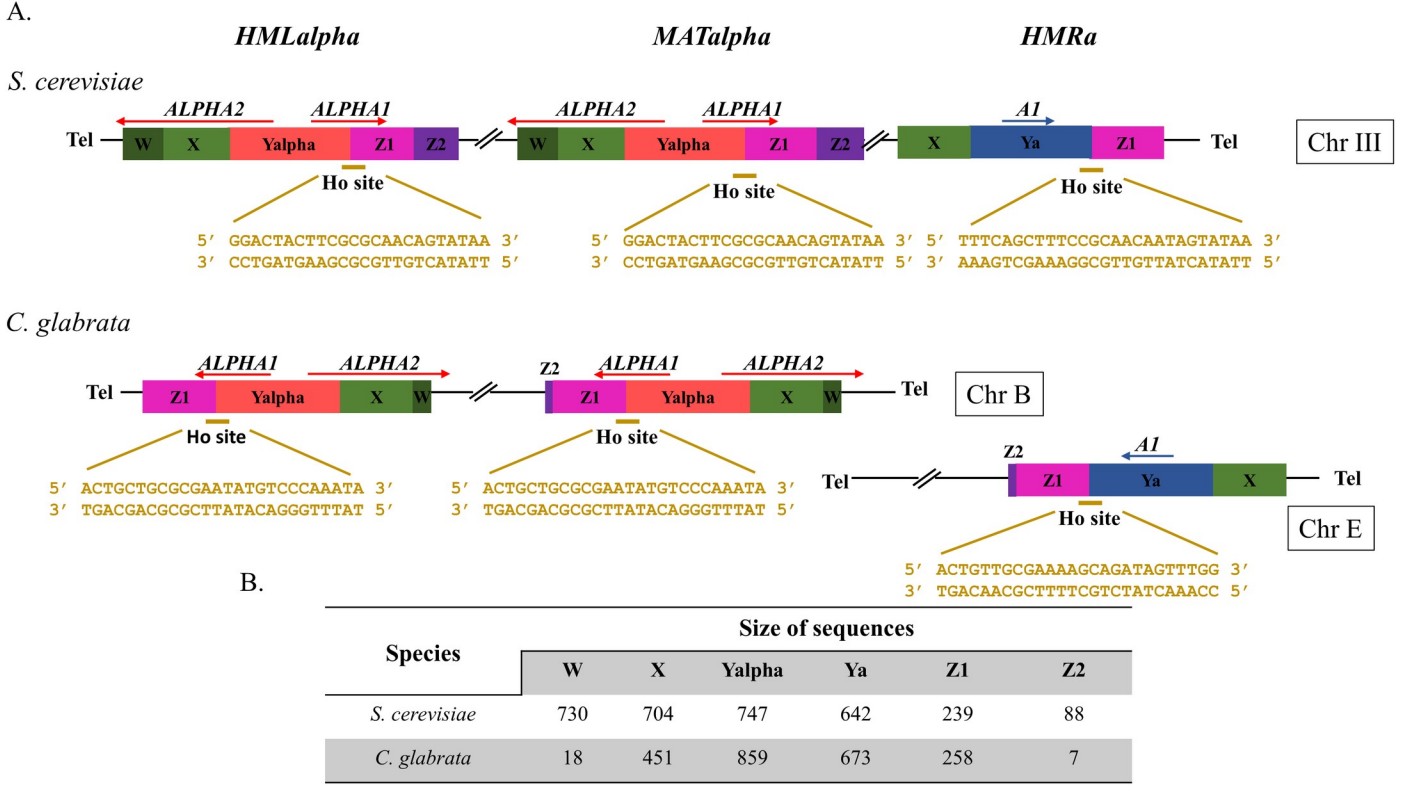

**Fig 1. Arrangement of *HML*, *MAT* and *HMR* in *S. cerevisiae* and in *C. glabrata* and size of sequences. (A)** Arrangement of the *HML*, *MAT*, *HMR* loci in *S. cerevisiae* and in *C. glabrata*. The Ho site sequence of each locus and species is shown in yellow. **(B)** Size of sequences forming the *HML*, *MAT*, *HMR* loci in *S. cerevisiae* and in *C. glabrata*.

thus have to be efficiently repaired to ensure cell viability. This can be achieved through two major pathways, non-homologous end-joining (NHEJ) and homologous recombination (HR) in the presence of a repair template. The DSB at *MAT* is repaired ~90% of the time by HR [11], probably because of efficient resection of the DSB that has been shown to prevent NHEJ [13]. The Ho cut at the *MAT* locus generates 4 bp, 3′-overhanging ends and its repair involves the following steps: the DSB ends are processed by several 5′ to 3′ exonucleases to create long 3′-ended tails [14]; single-strand tails are then converted to Rad51-coated nucleoprotein filaments, which search for homology and promote homologous template invasion [11]; once the homologous donor is found, the *MAT* locus is repaired by gene conversion. The homologous donor is one of the two silent loci located on the same chromosome as *MAT*: *HML* carrying the "alpha" information or *HMR* carrying the "a" information. The "alpha" or "a" sequence from *HML* or *HMR* respectively, replaces the original Y *MAT* sequence whereas *HML* and *HMR* remain unchanged. Despite the fact that *HML* and *HMR* contain the Ho recognition site, both are resistant to Ho cleavage, being located in heterochromatic regions [15]. It must be noted that the Ho recognition site is quite different between the "a" and "alpha" versions, one side is unchanged since it is located in the identical Z sequence, while the other side, corresponding to the end of the Y fragment, is different between the two mating-types ([Fig 1]). There is no measurable difference in the efficacy of the cut between the two sequences, nor between efficiency of mating-type switching from one to another [16,17]. This illustrates the fact that Ho is part of the family of meganucleases, that do not function like type II restriction endonucleases, but recognize large (larger than 12bp), degenerate, non-palindromic cut-sites.

In *S. cerevisiae*, a "donor preference" mechanism ensures an efficient mating-type switching at *MAT* by promoting the use of the silent locus from the opposite mating-type (*MATa* is preferentially repaired by *HMLalpha* and *MATalpha* by *HMRa*). This donor preference depends on both the "a" or "alpha" information at the *MAT* locus and the presence of a specific sequence, the recombination enhancer (RE), located between *HML* and *MAT* [18]. It must be noted that mating-type switching occurs only once per cell-cycle, in G1, and that this is thought to be regulated through the control of the expression of *HO* [12], but experiments of overexpression of *HO* under the control of a galactose-inducible promoter have shown that the switch can be induced in any part of the cell cycle [19]. Intriguingly, there is no report of switching back and forth between the two mating-types in such overexpression experiments, leaving open the possibility that another mechanism than *HO* gene expression, is responsible for the "unswitchability" of newly (i.e., unreplicated) switched cells.

*C. glabrata* is an opportunistic pathogenic yeast, phylogenetically close to *S. cerevisiae* [20]. Its genome has retained the three-locus system, with homologs of *HML*, *MATa/alpha*, and *HMR*, called <u>M</u>ating-<u>T</u>ype <u>L</u>ike (*MTL*) loci (Fig 1). The three loci display a structure comparable to *S. cerevisiae's*, the main difference being that *HMR* is located on a different chromosome from *HML* and *MAT* [20]. Despite these similarities, added to the fact that both *MATa* and *MATalpha* cells are found naturally and that they maintain some mating-type identity [21–23], *C. glabrata* is unable to switch mating-type spontaneously at an efficient level, even though rare signs of mating-type switching are observed in culture [24] and in populations [25]. We have previously shown that the expression of the *HO* gene from *S. cerevisiae* can trigger mating-type switching in *C. glabrata*, and that over 99% of *C. glabrata* cells are unable to survive the expression of *S. cerevisiae's* Ho (ScHo) [26]. We have also observed gene conversion events at the *HML* locus in survivors, revealing that, unlike *S. cerevisiae*, *HML* is not protected from the Ho cut. We suggested that the lethality was due to multiple chromosomal DSBs, which would prevent homologous recombination with an intact template in most cells.

In this work, we investigate the reasons for the lethality associated with mating-type switching induced by *Sc*Ho. For this purpose, we constructed a series of inconvertible (*Inc*) *C. glabrata* strains, mutated precisely at the Ho recognition site, allowing us to control the number and position of DNA breaks during induction of *Sc*Ho, as well as to track which donor sequence is used as template. We analyzed two aspects: viability, that reflects both the efficiency of the cut and the success of repair; and molecular structure of repaired loci, in wild-type and mutant strains, in order to reveal which repair pathways were used. We now show that *HMR* is also cut by Ho in wild-type strains of *C. glabrata*. In addition, by mimicking *S. cerevisiae's* situation, in which *HML* and *HMR* are protected from the cut, we unexpectedly find that one DSB at the *MAT* locus is sufficient to induce cell death, whereas one at *HML* or *HMR* is not. Finally, the use of the CRISPR-Cas9 technology enables us to show that mating-type switching can be induced independently of the Ho protein in *C. glabrata*, and that such switching is efficient and not lethal. Thus, we show for the first time that a chromosomal DSB is repaired by HR efficiently in *C. glabrata*, at *HML* and *HMR* (Ho-cut) and at *MAT* (Cas9-cut), indicating that, in principle, *MAT* switching could occur in this species. The fact that an Ho endonuclease, able to induce switching, also induces cell death may be evidence for degeneration of the three *MTLs*/Ho system in this species, in accordance with the observed asexuality.

## Results

### All three sites are cut by Ho in *C. glabrata*, including the one at *HMR*

We expressed *S. cerevisiae's* *HO* gene (*ScHO*) using the *URA3* selectable plasmid p7.1 in which *ScHO* is under control of the inducible *MET3* promoter [26]. As previously described,

**Table 1. Strains used in this work.**

| Strains | Parent | Genotype | Reference |
|---|---|---|---|
| *C. glabrata* **strains with wild-type Ho sites** | | | |
| CBS138 | | *HMLalpha MATalpha HMRa* | [27] |
| BG2 | | *HMLalpha MATa HMRa* | [28] |
| BG14 | BG2 | *HMalpha MATa HMRa ura3Δ::Tn903 G418^R* | [28] |
| BG87 | BG14 | *HMLalpha MATa HMRa ura3::Neo^R his3Δ* | [29] |
| HM100 | CBS138 | *HMLalpha MATalpha HMRa ura3Δ::KANMX* | [21] |
| HM100 *Δrad51* | HM100 | *HMLalpha MATalpha HMRa Δrad51 ura3Δ::KANMX* | This work. |
| *C. glabrata* **strains with mutated Ho sites** | | | |
| YL01 | HM100 | *HMLalpha MATalpha HMRa-inc ura3Δ::KANMX* | This work. |
| YL02 | HM100 | *HMLalpha-inc MATalpha HMRa ura3Δ::KANMX* | This work. |
| YL03-MATalpha | YL02 | *HMLalpha-inc MATalpha HMRa-inc ura3Δ::KANMX* | This work. |
| YL03-MATa | YL03-MATalpha | *HMLalpha-inc MATa HMRa-inc ura3Δ::KANMX* | This work. |
| YL04 | YL07 | *HMLalpha MATalpha-inc HMRa ura3Δ::KANMX* | This work. |
| YL05 | YL09 | *HMLalpha MATa-inc HMRa-inc ura3Δ::KANMX* | This work. |
| YL07 | YL02 | *HMLalpha-inc MATalpha-inc HMRa ura3Δ::KANMX* | This work. |
| YL09 | YL01 | *HMLa-inc MATa-inc HMRa-inc ura3Δ::KANMX* | This work. |
| YL10 | YL07 | *HMLalpha-inc MATalpha-inc HMRalpha-inc ura3Δ::KANMX* | This work. |
| SL09 | BG87 | *HMLa-inc MATa-inc HMRa-inc ura3::Neo^R his3Δ* | This work. |
| SL0A | YL01 | *HMLalpha MATalpha-inc HMRa-inc ura3Δ::KANMX* | This work. |
| SL0B | YL04 | *HMLa-inc MATalpha-inc HMRa ura3Δ::KANMX* | This work. |
| *C. glabrata* **strains with mutated Ho sites and/or deletion(s) of *HML*, *MAT* or *HMR*** | | | |
| CGM460 | BG14 | *Δhml MATalpha HMRa ura3Δ::Tn903 G418^R* | [22] |
| CGM390 | BG14 | *HMLalpha Δmat HMRa ura3Δ::Tn903 G418^R* | [22] |
| SL-CG1 | CGM390 | *HMLalpha Δmat Δhmr ura3Δ::Tn903 G418^R* | This work. |
| CGM498 | BG14 | *Δhml Δmat HMRa ura3Δ::Tn903 G418^R* | [22] |
| SL01 | BG87 | *HMLalpha MATa Δhmr ura3::Neo^R his3Δ* | This work. |
| SL-CG8 | SL01 | *HMLalpha-inc MATa Δhmr ura3::Neo^R his3Δ* | This work. |
| SL-CG9 | CGM460 | *Δhml MATa HMRalpha-inc ura3Δ::Tn903 G418^R* | This work. |
| SL-CG10 | SL01 | *HMLa-inc MATa-inc Δhmr ura3::Neo^R his3Δ* | This work. |
| SL-CG12 | CGM390 | *HMLalpha-inc Δmat HMRalpha-inc ura3Δ::Tn903 G418^R* | This work. |
| SL-CG14 | CGM460 | *Δhml MATalpha-inc HMRalpha-inc ura3Δ::Tn903 G418^R* | This work. |

expression of *ScHO* in wild-type strains of *C. glabrata*, leads to the death of about 99.9% of cells and we found that both *MAT* and *HML* are efficiently cut [26]. We further analyzed surviving colonies of HM100 (*HMLalpha MATalpha HMRa*, Table 1) by determining the mating-type at each *MTL* locus by PCR and we found that nearly all present switching at *HMR*, indicative of cutting (Table 2).

In order to formally confirm that mating-type switching at each *MTL* depends on HR, we inactivated *RAD51* (*CAGL0I05544g*) in the wild-type strain HM100 (Table 1). Inducing the Ho DSB in this strain leads to an even higher lethality than in the wild-type strain (Fig 2), and no mating-type switching is detected at any *MTL* locus (Table 2), confirming that switching relies on HR in *C. glabrata*.

As we hypothesized in our previous work [26], Ho-induced lethality in *C. glabrata* could be due to concomitant induction of multiple DSBs, in contrast to the situation in *S. cerevisiae* where *HML* and *HMR* are protected from the cut. These unrepairable cuts would lead to death by cell cycle arrest, or because cut and possibly degraded chromosomes segregating into daughter cells lack essential genes. Alternatively, we had mentioned the possibility that

**Table 2. Molecular structure of *MTLs* in surviving colonies after *Sc*Ho induction.**

| Strain | Locus screened | PCR results | Percentage of switch |
|---|---|---|---|
| *C. glabrata* strains with wild-type Ho sites | | | |
| HM100 wt (*HMLalpha MATalpha HMRa*) | *HML* | 25/34 mixed *HMLalpha/a* <br> 9/34 pure *HMLalpha* | 74 ± 6% |
| | *MAT* | 26/34 mixed *MATalpha/a* <br> 8/34 pure *MATalpha* | 76 ± 6% |
| | *HMR* | 30/34 mixed *HMRa/alpha* <br> 2/34 pure *HMRalpha* <br> 2/34 pure *HMRa* | 94 ± 6% |
| HM100 Δ*rad51* (*HMLalpha MATalpha HMRa*) | *HML* | 39/39 pure *HMLalpha* | 0% |
| | *MAT* | 39/39 pure *MATalpha* | 0% |
| | *HMR* | 39/39 pure *HMRa* | 0% |
| *C. glabrata* strains in which *MAT* is protected from the Ho cut | | | |
| YL05 (*HMLalpha MATa-inc HMRa-inc*) | *HML* | 60/60 pure *HMLa-inc* | 100 ± 8% |
| YL07 (*HMlalpha-inc MATalpha-inc HMRa*) | *HMR* | 34/36 mixed *HMRa/alpha-inc* <br> 2/36 pure *HMRalpha-inc* | 100 ± 6% |
| *C. glabrata* strains in which *MAT* can undergo the Ho cut | | | |
| SL-CG8 (*HMLalpha-inc MATa Δhmr*) | *MAT* | 36/36 pure *MATalpha-inc* | 100 ± 6% |
| SL-CG9 (*Δhml MATa HMRalpha-inc*) | *MAT* | 36/36 pure *MATalpha-inc* | 100 ± 6% |

Colonies surviving *Sc*Ho induction are screened by PCR at each locus that can be cut by *Sc*Ho. Percentage of switch is calculated as the ratio of the total number of pure and mixed colonies exhibiting mating-type switching divided by the total number of surviving colonies tested, expressed as percentage. The square root of the number of surviving colonies screened is used as standard error in last column.

switching leads to the repair of the Ho-cut locus by an intact Ho-site containing locus would cause never-ending cycles of cutting and repair that could also lead to cell death in our conditions of continuous induction on plates. We had dismissed this eventuality as unlikely, since the percentage of lethality and switched survivors is highly reproducible in our experiments. Even though the situation is the same in *S. cerevisiae* when overexpressing Ho with a galactose-inducible promoter, and no switching back and forth between the two mating-types has been reported in *S. cerevisiae* [14,19], we had no experimental proof that this did not happen in *C. glabrata*, in our experimental conditions. We thus decided to design experiments which would address both the question of the number of cut loci, and the question of the number of cuts per locus. For this, as explained below, we have used strains containing inconvertible Ho sites (*Inc*) and/or deletions of *MTLs* and we have performed a time-course experiment of induction (see Material and methods).

## Cleavage at *HML* and *HMR* is not an important contributor to lethality

As soon as we started performing experiments using strains with inconvertible *MTLs*, we noticed difference of behavior between the *MAT* locus and *HML* and *HMR*. We first used strains YL05 (*HMLalpha MATa-inc HMRa-inc)* and YL07 (*HMLalpha-inc MATalpha-inc HMRa*), Table 1). In these strains, we mutated the Ho sites in the region known to be essential for Ho cutting in *S. cerevisiae* [30], yielding loci *MAT-inc*, *HML-inc*, *HMR-inc* (S1 Fig), in configurations where only either the *HML* or *HMR* locus can be cut and repaired by the two non-cleavable donors *MAT-inc*, and *HMR-inc* or *HML-inc*.

Upon Ho induction, cell viability drastically increases to 35 to 55%, depending on the strain (Fig 2). Survival does not reach 100% but is more than 2.700 times higher than in the wild-type isogenic strain, HM100 (HM100 vs YL05 P-value$<10^{-4}$ and HM100 vs YL07 P-value$<10^{-9}$, Wilcoxon tests) (Fig 2). We then expressed *HO* in a strain in which only *MAT* is protected

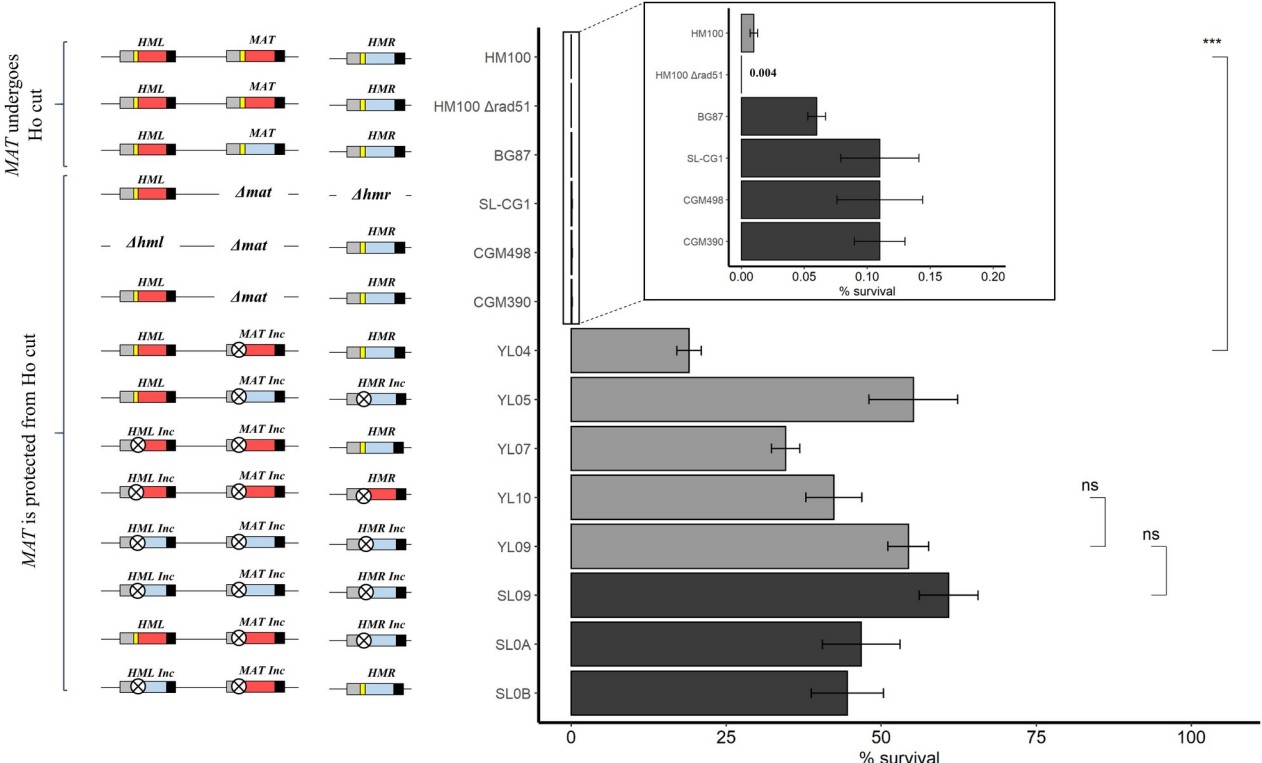

**Fig 2. Survival to *Sc*Ho induction of strains bearing combinations of wild-type and/or mutated Ho sites.** On the left, diagram of *MTL* configuration of strains is shown with the corresponding survival histogram to *Sc*Ho induction on the right. The blue box represents the Ya, the red box Yalpha, the yellow bar wild-type Ho site and the crossed circle mutated Ho site (*Inc* loci) (not to scale). On the histogram, black bars are for strains from the BG87 background, grey bars are for strains from the HM100 background. Results for strains HM100 and BG87 are from (26). Values from, at least, four experiments were averaged, the SEM used as estimate of the error and the P-value was calculated using the Wilcoxon test. ***: P-value<0.001. ns: non-significant.

from the cut, while both *HML* and *HMR* can be cleaved by Ho (Strain YL04 *HMLalpha MATalpha-inc HMRa*, Table 1). In this strain, cell viability reaches ~20% which is 2.000 times higher than in the wild-type isogenic strain, HM100 (P-value<0.001, Wilcoxon test) (Fig 2).

We analyzed the molecular structure of the *HML* and *HMR* loci in surviving colonies of strains YL05 and YL07 respectively, by PCR using primers specific of the mating-type carried by the *MTLs* ("alpha" or"a", wt or inc, S1 Table and S2 Fig). This allows the distinction of the original *HML* or *HMR* locus from the repaired locus that has become resistant to cutting. We found that 100% of surviving colonies showed mating-type switching of *HML* or *HMR* (Table 2). As mating-type switching reflects the efficiency of the Ho-cut, this suggests that both *HML* and *HMR* are efficiently cut by *Sc*Ho and that, even though we found some mixed colonies for *HMR* in strain YL07, *MTLs* they are repaired by HR.

In order to confirm this, we induced expression of *Sc*Ho in strains in which either *HML* or *HMR* can be cut by Ho in absence of any repair template, the two other loci being completely deleted (strain SL-CG1, *HMLalpha Δmat Δhmr* and strain CGM498, *Δhml Δmat HMRa*, Table 1) (Fig 2). Upon Ho induction, we found that survival rate does not exceed 0.2% in both strains, suggesting that the Ho-cut is efficient at both *HML* and *HMR* and that the Ho-DSB at *HML* and *HMR* causes lethality when it cannot be repaired by HR. To explore this further, we induced *Sc*Ho in a strain in which both *HML* and *HMR* can be cut and *MAT* is deleted (strain CGM390 *HMLalpha Δmat HMRa*, Table 1) (Fig 2). Once again, survival does not exceed 0.2%, suggesting that *HML* and *HMR* are cut concomitantly and thus cannot serve as templates for

one another. This reinforces the hypothesis that simultaneous DSBs, happening in wild-type strains, participate in the high lethality observed.

Altogether, these results suggest that the efficient Ho-cut at *HML* and *HMR* is not an important contributor to lethality in all configurations where they can be repaired by HR. In the absence of HR, no other mechanism such as NHEJ is able to take over the repair of the Ho-cut, and thus cell survival remains low. Results also show that protecting the *MAT* locus from the Ho-cut significantly increases survival. This is also confirmed by the fact that strains containing no Ho-site at any *MTL* locus survive the Ho induction at around 50% (strains YL10, YL09 and SL09, Table 1) (Fig 2).

## A single *Sc*Ho-DSB at *MAT* is sufficient to induce cell death in *C. glabrata*

In order to measure the impact of the Ho cleavage at *MAT* on cell survival, we mimicked the situation in *S. cerevisiae*, where a single recipient of the Ho-induced DSB, the *MAT* locus, can be repaired by the two non-cleavable donors *HML* and *HMR*, i.e. strains YL03-MATalpha (*HMLalpha-inc MATalpha HMRa-inc*) and YL03-MATa (*HMLalpha-inc MATa HMRa-inc*) (Table 1). Expression of *ScHO* in those strains leads to a lethality similar to the one obtained in wild-type strains HM100 and BG87 (Fig 3) and all surviving colonies have switched, whatever the mating-type at *MAT* (Table 2).

Thus, a single Ho-induced DSB at *MAT*, whatever its mating-type, is sufficient to induce massive cell death in *C. glabrata*. Furthermore, this experiment allows us to reconsider the question of lethality due to never-ending cycles of cutting and repair. Indeed, if this was the reason for mortality of cells when cutting at *MAT*, then, since repairing with an *Inc* locus leads to an unswitchable locus, the mortality should be decreased in these strains upon induction.

## Lethality is not due to toxic recombinational repair intermediates

Since we know that the mating-type switching system in *C. glabrata* is not very efficient, it is legitimate to wonder whether the degeneration of the mating-type switching mechanism

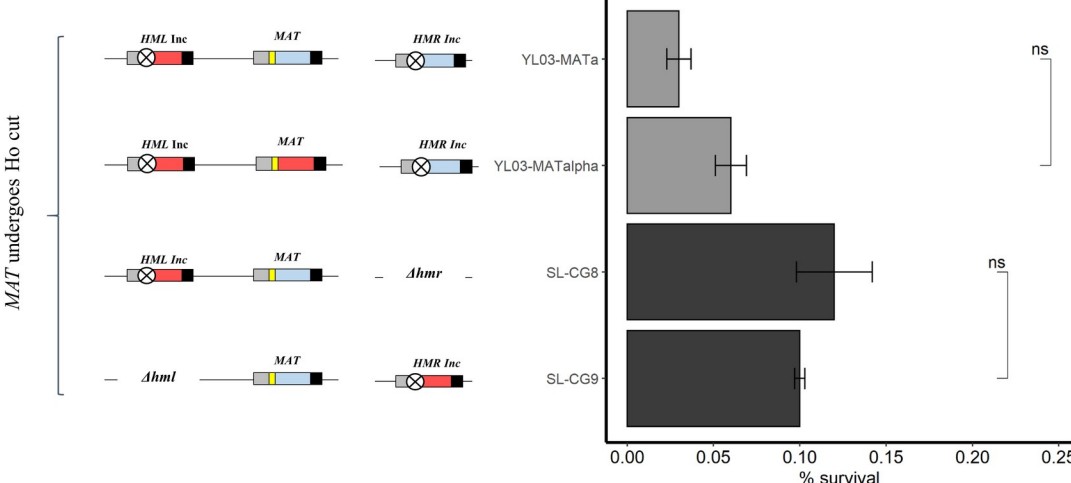

**Fig 3. Survival to *Sc*Ho induction of strains in which only the *MAT* locus can be cut.** On the left, diagram of *MTL* configuration of strains is shown with the corresponding survival histogram to *Sc*Ho induction on the right. The blue box represents the Ya, the red box Yalpha, the yellow bar wild-type Ho site and the crossed circle mutated Ho site (*Inc* loci) (not to scale). On the histogram, black bars are for strains from the BG87 background, grey bars are for strains from the HM100 background. Values from, at least, four experiments were averaged, the SEM used as estimate of the error and the P-value was calculated using the Wilcoxon test. ns: non-significant.

could lead to abnormal HR intermediates. We asked whether such repair intermediates could be toxic and cause death. For example, the two ends of the broken *MAT* locus could invade both *HML* and *HMR*, leading to non-resolvable structures. The fact that *HMR* is not on the same chromosome as *HML* and *MAT* could be an additional problem, if, for example, repair of *MAT* occurs principally with *HMR* and this leads to lethal rearrangements.

In order to test this, we constructed two strains in which *MAT* can be cut by Ho and can only be repaired either by *HML* or by *HMR* (SL-CG8, *HMLalpha-inc MATa Δhmr*, and SL-CG9, *Δhml MATa HMRalpha-inc*, respectively, Table 1). Expression of *ScHO* in both strains leads to a high lethality (Fig 3), similar to the ones of the wild-type or YL03 strains (*HMLalpha-inc MATalpha or MATa HMRa-inc*, Table 1). We analyzed the molecular structure in surviving colonies (Table 2). All have switched, whatever the location of the repair template (*HML* in strain SL-CG8 and *HMR* in strain SL-CG9).

Thus, forcing *MAT* to repair solely on *HML* or *HMR* results in the same lethality as in wild-type strains. From this, we conclude that the cut *MAT* locus DNA ends probably do not interact with both *HML* and *HMR* in such a way that it is toxic to cells, and that there is no specific problem due to the fact that *HMR* is on another chromosome than *MAT*. It, therefore, seems unlikely that lethality could be due to non-resolvable HR intermediates.

## Time course experiments reveal that growth arrest is quick and irreversible leading to cell death

In order to shed light on whether the toxic effect leading to cell death could be reversible and if not, whether the effect operates rapidly or not, we performed a time course experiment in which Ho is induced in liquid medium and its expression is repressed, at different time points, by plating cells on repressive medium. The survival can thus be calculated by the ratio of colonies obtained on repressive medium to the theoretical number of cells plated.

In order to easily follow events, we used again strain SL-CG9 (*Δhml MATa HMRalpha-inc*, Table 1), where only the *MAT* locus can be cut and a single inconvertible template for HR is present. This allows to follow a single Ho-cut and a single repair event at the *MAT* locus. We scored the survival along the time-course experiment as well as the percentage of survivors that have undergone switching. As shown on Fig 4, two hours upon induction of a single Ho-

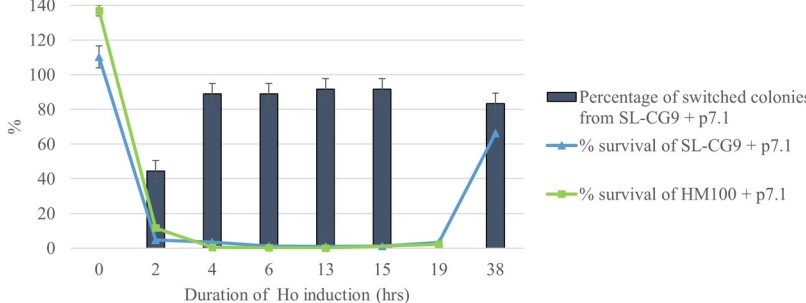

**Fig 4. Time-course experiment of *Sc*Ho induction in liquid culture.** Strains SL-CG9 (blue curve) and HM100 (green curve) were followed during a time-course experiment of induction of *Sc*Ho in liquid culture. Survival is shown on the Y-axis as a curve and is calculated by comparing the number of colony-forming units on SC-Rep with the number of cells plated, as estimated by counting. This is normalized by dividing it by the survival rate of the control strain, i.e., the strain transformed by pYR32, grown in the same conditions. Values from four experiments were averaged and the SEM is used as an estimate of the error. Percentage of switched colonies is also shown for strain SL-CG9 as a histogram. This is expressed as a percentage of colonies showing switching by PCR at *MAT* on the total number of colonies screened (this was not performed at T = 19hrs). At each time-point, except T = 19hrs, 36 colonies over the four experiments were screened and the square root of 36 is used as an estimate of the error.

DSB at *MAT*, survival drastically drops to less than 2%. From T = 4 hrs to T = 15 hrs, survival remains very low. Molecular analysis shows that mating-type switching in survivors reaches its maximum very rapidly after four hours of induction, at around 90%. All screened colonies display a pure genotype in PCR just as on solid medium (see Table 2). The last two points of our experiment, T = 19 hrs and T = 38 hrs show that survivors have invaded the liquid culture, thus giving rise to many colonies on repressive medium plates. These survivors thus consist of 90% of switched inconvertible clones, and around 10% of cells that have escaped Ho-induction probably by rearranging or mutating the plasmid. These results show that the toxic effect of inducing an Ho-cut at the *MAT* locus is irreversible even after only two hours in induction medium.

We next wanted to check whether mortality was as quick and as irreversible in wild-type strains. Induction in HM100 leads to the same pattern as SL-CG9 (Fig 4). This confirms that the lethality in HM100 is not due to never-ending cycles of cutting and repair since it cannot happen in SL-CG9 and the lethality occurs at the same rate.

## The lethality induced by the DSB at *MAT* is specific to *Sc*Ho

We wanted to investigate whether the lethality is caused by the DSB at *MAT per se* or by the specific combination of *MAT* with *Sc*Ho. Since we have previously shown that inducing other Ho endonucleases from the *Nakaseomyces*, including *C. glabrata*'s own gene, does not result in high lethality nor efficient mating-type switching [26], we decided to use the CRISPR-Cas9 system from [31]. This system relies on a unique *URA3* plasmid, pCGLM1, in which the *CAS9* gene is under the control of the inducible *MET3* promoter, as is the case for the *HO* gene in the p7.1 plasmid used in the experiments above. This allows us to induce a DSB at *MAT* with Cas9, in the same conditions as with *Sc*Ho.

In order to allow a full comparison between *Sc*Ho and Cas9 induction, we wanted to generate a single DSB at the same locus, here *MAT*, whatever the endonuclease used. Indeed, the Ho site between *MTLa* and *MTLalpha* is very different while the Cas9-cut is directed with a specific gRNA, we used SL-CG8 and SL-CG9 (*HMLalpha-inc MATa Δhmr*, and *Δhml MATa HMRalpha-inc*, respectively, Table 1) with a gRNA that targets only the *MAT* locus, since it is directed to the Ya sequence (Fig 5A). This gRNA allows to target the Cas9-cut from 14 pb from the Ho cut (Fig 5A).

Fig 5B shows that induction of the Cas9-cut in both SL-CG8 and SL-CG9 does not lead to any lethality. We wondered whether Cas9 had indeed cut the *MAT* locus by screening mating-type switching of surviving colonies by PCR (Table 3).

Results show that the Cas9-induced DSB leads to mating-type switching as efficiently as the Ho protein (Table 2 and Table 3). Indeed, 72 to 85% of the colonies tested presented "alpha-inc" information at *MAT*, confirming the cut of this locus by Cas9 and induction of mating-type switching (Table 3). This gRNA is adjacent to an optimal NGG PAM sequence and targets a site very close to the Ho site, 14 pb away from the Ho cut.

We then asked whether we could detect a transient lethality, reflecting the Cas9-cut. Indeed, previous experiments, using the same system for inducing a Cas9-cut, but at the unrelated *ADE2* locus, have shown that a transient lethality occurred in liquid culture in a time-course experiment, whereas no apparent lethality was detected on induction plates [31]. Thus, we performed a time-course experiment with Cas9 in strain SL-CG9. Fig 5D shows that Cas9 induction never leads to a sharp increase in lethality at any time point and survival remains between 65 and 92% for the whole experiment. Surprisingly, contrary to what we observe in induction on plates, screening of mating-type switching at *MAT* reveals that only ~20 to 36% of surviving colonies have switched mating-type (see Discussion).

These results show for the first time that mating-type switching can be induced without any lethality in *C. glabrata* using the CRISPR-Cas9 system. In conclusion, the fact that the *MAT*

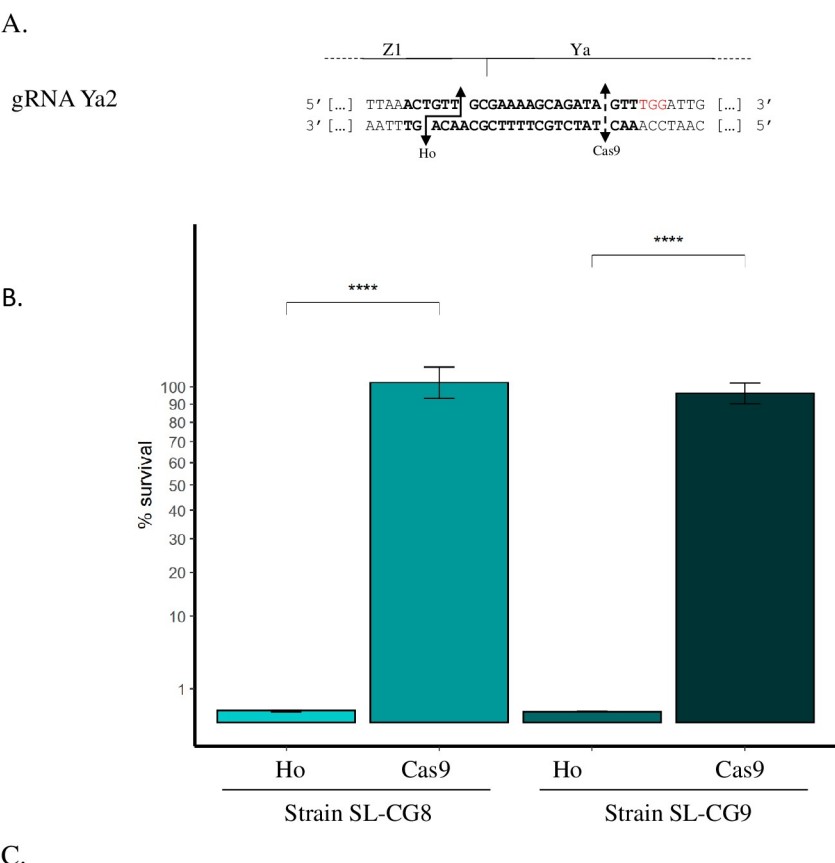

**A.**

gRNA Ya2

**B.**

**C.**

**Fig 5. Survival upon Cas9 induction and gRNAs used. (A)** gRNA Ya2 targeting the *MATa* locus of *C. glabrata*. Sequence shown is a segment of the *MATa* locus of BG87, including the gRNA in bold and the PAM sequence in red. Plain double arrow indicates the Ho cleavage site and dashed double arrow the Cas9 cleavage site. **(B)** Survival of strains SL-CG8 and SL-CG9 upon Cas9-induced DSB at *MAT*. Induction is performed on solid medium. Results for strains SL-CG8 and SL-CG9 upon *Sc*Ho induction are from Fig 3. Values from four experiments were averaged, the SEM used as estimate of the error and the P-value was calculated using the Wilcoxon test. ***: P-value<0.001. **(C)** Induction was performed in liquid during a time-course experiment for strain SL-CG9 expressing Cas9 (harboring pCGLM1-Ya2). The Y-axis represents both the survival (curve) expressed as a percentage, and the percentage of switched colonies (histogram). Survival is calculated by comparing the number of colony-forming units on SC-Rep with the number of cells plated, as estimated by counting; and is normalized by dividing it by the survival rate of the control strain, i.e., strain SL-CG9 transformed by pCGLM1 for Cas9 induction, grown in the same conditions. For survival rate, values from four experiments were averaged and the SEM is used as estimate of the error. For the percentage of switched colonies, the square root of the number of surviving colonies screened is used, i.e., sqrt of 36. For time-course experiments, at points T = 17 and T = 21 hrs, no PCR of surviving colonies was performed.

**Table 3. Molecular structure of the *MAT* locus in colonies after Cas9 induction.**

| Strain | Locus screened | gRNA used | PCR results of surviving colonies | PCR results of sub-clones | Percentage of switch |
|---|---|---|---|---|---|
| SL-CG8 (*HMLalpha-inc MATa Δhmr*) | *MAT* | Ya2 | 36/40 mixed *MATalpha-inc/a*<br>4/40 pure *MATalpha-inc* | 40/48 *MATalpha-inc*<br>8/48 *MATa* | 85% |
| SL-CG9 (*Δhml MATa HMRalpha-inc*) | *MAT* | Ya2 | 32/45 mixed *MATalpha-inc/a*<br>7/45 pure *MATalpha-inc*<br>6/45 pure *MATa* | 38/48 *MATalpha-inc*<br>10/48 *MATa* | 72% |

After Cas9 induction, colonies are screened by PCR at *MAT*. When PCR revealed mixed colonies, at least four individual colonies are sub-cloned in order to get the ratio of each mating-type; percentage of switch is calculated as follows: the ratio of the number of pure switched colonies on the total number is added to the ratio of the number of mixed colonies on the total number screened multiplied by the ratio of the number of pure switched sub-clones on the total number of sub-clones screened, expressed as percentage. An example for the molecular analysis of SL-CG8 using gRNA Ya2, is: (4/40 + (36/40 x 40/48)) x 100 = 85%.

locus can be cut and repaired by HR without any accompanying high lethality demonstrates that it is *Sc*Ho cutting specifically at the *MAT* locus that is highly lethal in *C. glabrata*.

## Choice of repair template reveals a complex interplay between the *MTL* loci

As the three *MTL*s are efficiently cut by *Sc*Ho and as the cut at *MAT* is the only one to lead to a high lethality, we decided to study these differences in *C. glabrata* by asking how each locus interacts with the two other templates.

In wild-type strains of *S. cerevisiae*, only the *MAT* locus is cut, and the mechanism of mating-type switching is productive thanks to control by the sexual identity of the cell. Indeed, the use of the donor locus of the opposite mating-type to repair the DSB at *MAT* is promoted ("donor preference") [18]. Since it has been shown that sexual identity of cells is maintained in *C. glabrata*, at least in *MATa* cells (i.e., *MATa* cells express *MATa* identity, but *MATalpha* cells have no mating-type specific expression of key genes [21,22]), we asked whether this "donor preference" from *S. cerevisiae* is conserved in *C. glabrata* at the *MAT* locus, and also whether *HML* and *HMR* use a preferential template for repair. For this, we constructed strains that carry different and inconvertible mating-types in various combinations. First, for the *MAT* locus, we used strains YL03-MATalpha and YL03-MATa (*HMLalpha-inc MATalpha HMRa-inc* and *HMLalpha-inc MATa HMRa-inc*, respectively, Table 1). Analysis of surviving colonies from strain YL03-MATa shows that 78% display only the alpha-inc information at *MAT*, the remainder exhibiting various genotypes, pure a-inc for 3%, and mixed alpha-inc and a-inc for 19% (Table 4). The overwhelming percentage of pure colonies displaying only the alpha-inc information at *MAT* demonstrates that *HML* is the preferred template for repair of the *MATa* locus.

In strain YL03-MATalpha: 84% of tested surviving colonies display only the *MATalpha-inc* genotype, the others displaying various mixed or pure genotypes at *MAT* (Table 4). Thus, in contrast to *S. cerevisiae*, *HML* is preferentially used as template for repair in *C. glabrata*, whatever the mating-type at *MAT*.

In order to know by which template *HML* is preferentially repaired, we performed a molecular analysis of the *HML* locus, upon ScHo induction, in a strain that carries different and inconvertible mating-types at *MAT* and *HMR* (strain SL0A, *HMLalpha MATalpha-inc HMRa-inc*, Table 1). Analysis of surviving colonies shows that 47% display pure alpha-inc information, 40% are alpha-inc and a-inc in mixed colonies and 13% are pure a-inc (Table 4). This indicates that *HML* preferentially repairs the DSB using *MAT* over *HMR*.

In the same way, molecular analysis of the *HMR* locus was performed in strain SL0B (*HMLa-inc MATalpha-inc HMRa*, Table 1). Table 4 shows that, 9% of surviving colonies are pure a-inc and 84% display both a and a-inc information We observe no repair event where

**Table 4. Analysis of template choice for repair of Ho cut at *MTLs*.**

| Strain | Locus screened | PCR results | Use of each locus |
|---|---|---|---|
| YL03-MATa (*HMLalpha-inc MATa HMRa-inc*) | *MAT* | 25/32 pure *MATalpha-inc*<br>6/32 mixed *MATalpha-inc/a-inc*<br>1/32 pure *MATa-inc* | *HML*: 96%<br>*HMR*: 4% |
| YL03-MATalpha (*HMLalpha-inc MATalpha HMRa-inc*) | *MAT* | 42/50 pure *MATalpha-inc*<br>5/50 mixed *MATalpha-inc/a-inc*<br>2/50 pure *MATa-inc*<br>1/50 pure *MATalpha* | *HML*: 95%<br>*HMR*: 5% |
| SL0A (*HMLalpha MATalpha-inc HMRa-inc*) | *HML* | 33/69 pure *HMLalpha-inc*<br>20/69 mixed *HMLalpha-inc/a-inc*<br>9/69 pure *HMLa-inc* purs<br>7/69 mixed *HMLalpha/alpha-inc/a-inc* | *MAT*: 79%<br>*HMR*: 21% |
| SL0B (*HMLa-inc MATalpha-inc HMRa*) | *HMR* | 38/45 mixed *HMRa/a-inc*<br>4/45 pure *HMRa-inc*<br>3/45 pure *HMRa* | *HML*: ~100%<br>*MAT*: ~0% |

In this experiment, strains are chosen so that template choice for *Sc*Ho-cut can be analyzed. As before, colonies are screened by PCR at the single locus that can be cut by *Sc*Ho. In the first three strains, since the percentage of pure colonies alone is sufficient to know which template is preferentially used for repair, the percentage of the use of each locus is calculated by only taking into account the number of pure switched surviving colonies (i.e., we did not sub-clone mixed colonies and we also omitted the pure unswitched colony obtained in strain YL03-MATalpha). The use of each locus is calculated as the ratio of the number of colonies showing the use of that locus on the total number of pure switched surviving colonies screened, expressed as percentage. In the case of strain SL0B, we fail to detect by PCR the use of *MATalpha-inc* to repair *HMR*, thus we estimate that in switched colonies, the use of *HML* is of the order of 100%.

*MATalpha-inc* was used as template. *HML* is thus the preferred template for the repair of *HMR*.

Overall, these results show that *MAT* and *HML* preferentially repair on each other and that *HMR* is preferentially repaired by *HML*.

## Exploring the residual lethality in the absence of *MTL*-cuts

When cuts are non-repairable by HR; i.e., the three strains containing wild-type Ho sites: HM100 *Δrad51* (*HMLalpha MATalpha HMRa*), SL-CG1 (*HMLalpha Δmat Δhmr*) and CGM498 (*Δhml Δmat HMRa*) and inexplicably, the YL03 strains (*HMLalpha-inc MATa or alpha HMRa-inc*) (Table 1), 99.9% of the cells die. As shown above, in other strains where the Ho-cut is repaired by HR (at *HML* and *HMR*) and in strains where there is no Ho cut (Y09, YL10 and SL09, Table 1) survival varies between 20% and 61%, never reaching 100%. This residual lethality can be explained by a general toxic effect of the expression of a heterologous protein in *C. glabrata* or by of the existence of (a) cryptic unrepairable Ho site(s) elsewhere in the genome. We thus decided to test this residual lethality in other strains where no cutting occurs. For this, we constructed strains where one *MTL* is deleted and two are inconvertible, SL-CG10 (*HMLa-inc MATa-inc Δhmr*), SL-CG12 (*HMLalpha-inc Δmat HMRalpha-inc*) and SL-CG14 (*Δhml MATalpha-inc HMRalpha-inc*) (Table 1). Upon Ho induction, survival in these three strains is around ~80%, whatever the *MTL* deleted (Fig 6). Comparing the survival of SL-CG10, SL-CG12 and SL-CG14 to SL09 (*HMLa-inc MATa-inc HMRa-inc*) (same genetic background and highest survival in our previous experiment, i.e., 60%) show that the increase in survival is significant (P-value<0.05 in each pairwise combination of SL09 with SL-CG10, SL-CG12 and SL-CG14, Wilcoxon tests). If the residual lethality observed was due to a general toxic effect or to (an) extra Ho site(s), the lethality would be the same in those strains as in strains that are inconvertible for the three *MTLs* (YL10, YL09 and SL09) (Fig 3) since the constructions do not modify the context outside of the *MTL* loci.

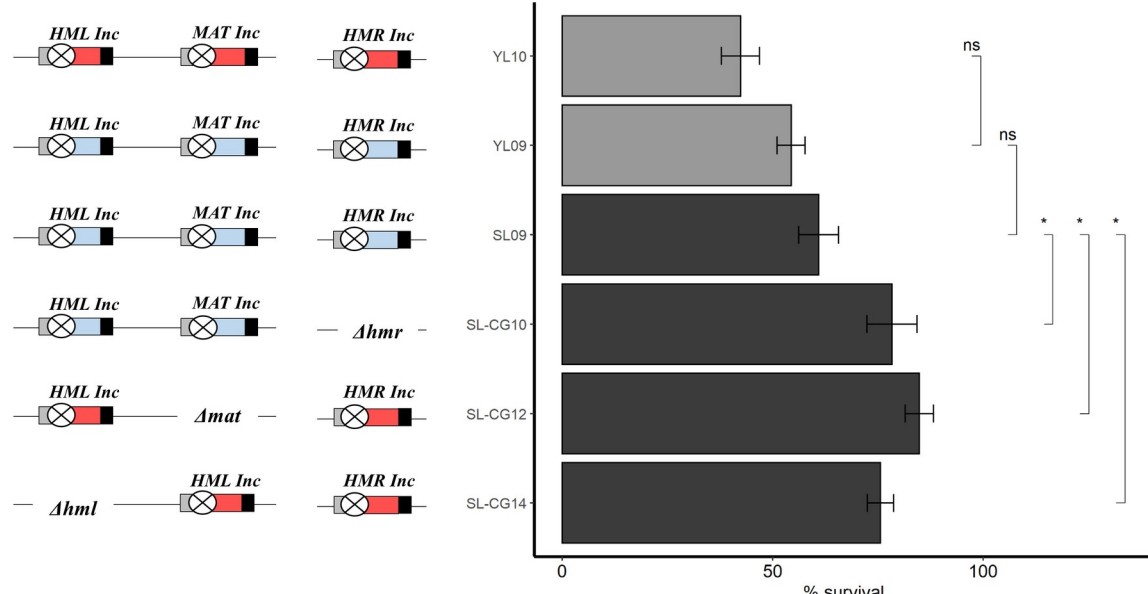

**Fig 6. Survival of strains without wild-type Ho sites and containing deletion of one *MTL*.** Blue box represents Ya, red box Yalpha and crossed circle mutated Ho site (*Inc* loci) (not to scale). Histogram shows survival of strains with corresponding *MTL* configuration. Results for strains YL10, YL09 and SL09 are from Fig 2. Values from, at least, four experiments were averaged, the SEM used as estimate of the error and the P-value was calculated using the Wilcoxon test. *: P-value<0.05. ns: non-significant.

Since the only difference between those strains is the number of inconvertible sites and thus potential binding sites for *Sc*Ho, we believe this may explain residual lethality by binding of *Sc*Ho (see Discussion).

## Discussion

Mating-type switching is a highly regulated mechanism that relies on a chromosomal DSB. DSBs are a major threat for genome integrity [32]. Repair of such damage is thus essential and can be achieved through Rad51-dependent HR which involves many steps in order to succeed: search for homology involving Rad51 and Rad52 in *S. cerevisiae*, copy on the donor locus and displacement and resolution of the double Holliday junction [33]. In *S. cerevisiae*, the DSB at the *MAT* locus is repaired by HR using *HMR* or *HML* as template, depending on the original mating-type of the cell. *C. glabrata* does not switch mating types spontaneously at high frequency [24]. We have previously shown that overexpression of *HO* genes from *C. glabrata* and related species fails to induce efficient mating-type switching and that switching can be efficiently induced by expressing *ScHO*, but that it is lethal to most cells [26]. Our previous work also showed that the *HML* locus is cut in *C. glabrata*; something that never happens in wild-type strains of *S. cerevisiae* [26]. In this work, we aimed at understanding the link between mating-type switching and cell death in *C. glabrata*. First, we show that *HMR* is also cut in wild-type strains of *C. glabrata*, overexpressing *ScHO*. Next, we constructed strains with mutated uncuttable Ho sites (inconvertible, *Inc*) and with deleted *MTL* loci, in order to examine survival to individual DSBs at the different *MTL* loci as well as knowing which *MTL* has been used as template for repair.

In *S. cerevisiae*, the donor preference mechanism ensures an efficient mating-type switching at *MAT* by promoting the use of the template from the opposite mating-type, in repair [18]. We found, in *C. glabrata*, that whatever the mating-type at *MAT*, *HML* is preferentially used as

template for repair. Thus, donor preference from *S. cerevisiae* seems not to be conserved in *C. glabrata*. In its absence, it also shows that the length of the sequence homology does not influence the use of the donor for repair. Indeed, the Ya and Yalpha segments, determining mating-type, share no homology and we observe that *MATa*, containing Ya, repairs preferentially using *HMLalpha*, containing Yalpha. We also demonstrate that *HML* is preferentially repaired using *MAT* and that *HMR* is preferentially repaired using *HML*, revealing a complex interplay between those different chromosomal segments. Loss of donor preference at *MAT*, along with the fact that the *C. glabrata* endogenous Ho protein fails to induce efficient mating-type switching [26], point to the possible degeneration of the mating-type switching system in *C. glabrata*. Thus, rules of DSB repair by HR observed in our inductions may reflect true preferences, independently of the *MAT*/Ho switching system. When the DSB can be repaired by HR using a template on the same chromosome or another chromosome, the intra-chromosomal template will be mostly chosen over the other (*HML* repairs preferentially with *MAT* and *vice versa*). When the DSB can only be repaired by an extra-chromosomal template, the sub-telomeric *HMR* was repaired preferentially using the sub-telomeric template *HML*. We can speculate that HR using a template on a same chromosome is preferred for DSB repair and in the absence of such a template, the sub-telomeric loci are repaired using other sub-telomeric loci.

In our previous work, we hypothesized that multiple DSBs at the *MTL* loci would be unrepairable and that this was the cause for lethality when mating-type switching is induced. As cited above, to mimic *S. cerevisiae*'s situation, in which *MAT* is the only recipient of the Ho cut, we mutated the Ho site at both *HML* and *HMR*. We are now able to demonstrate that one *Sc*Ho-DSB at the *MAT* locus is sufficient to induce cell death at a similar level to wild-type cells, thus invalidating our previous hypothesis. This means that, even in the presence of two intact homologous sequences, the *MAT* locus is not able to repair the break. More surprisingly, the DSB at *MAT* is only lethal when it is performed by the *Sc*Ho protein. We show, for the first time, that mating-type switching can be triggered efficiently by CRISPR-Cas9, thus independently of the Ho protein, in *C. glabrata*. This has been shown only recently in the model species *S. cerevisiae* [34]. No lethality is observed after a Cas9-DSB at *MAT* on plates. In liquid induction experiments, we observe a transient lethality of ~35% and a lower efficiency of switch (~30%). In both plate and liquid inductions, the Cas9 expression plasmid is constantly under selection pressure. We believe that observed discrepancies can be explained if induction is less efficient on individual cells in liquid medium than on plates, implying that cells in liquid medium can mutate the plasmid before switching (in the *CAS9* gene or its promoter in such way that *CAS9* is not expressed anymore, or in the gRNA sequence). These cells will never switch and can invade the culture. In contrast, some cells undergoing switching may not recover (~35% of lethality) but the ones that have survived the switch become "immune" to the Cas9-DSB, thus alleviating the need for mutating the Cas9 plasmid. Nonetheless, these cells may grow more slowly because of the maintenance of *CAS9*'s expression and are thus less likely to be in excess in the culture. Thus, these *MAT*-switched cells will be diluted and less represented on repressive plates.

To go back to the lethality induced by *Sc*Ho, unless the difference in the lethality with Cas9 is due to the 14 bp shift in cutting, which seems highly unlikely, these results suggest that the *Sc*Ho protein prevents DSB repair specifically at the *MAT* locus of *C. glabrata*, in such a way that 99.9% of the cells die. We have recently published that, in heterothallic strains of *N. delphensis*, overexpression of *Sc*Ho with the same plasmid as in this work leads to efficient switching, without lethality [35]. It is surprising that *Sc*Ho could have a deleterious effect in a locus- and species-specific manner. As in all three-loci-based mating-type switching systems, the three *MTL* loci of *C. glabrata* share identical sequences and only differ by the mating-type carried and/or their location in the genome [36]. We know that mating-type does not influence

lethality in any of our experiments. Thus, only the location of the *MAT* locus could explain the specificity of lethality induced by *Sc*Ho. The *MAT* locus is located in a central region on chromosome B whereas *HML* and *HMR* are positioned in sub-telomeric regions on chromosome B and E, respectively [36]. Thus, the *Sc*Ho specificity for *MAT* could only be achieved either through the structure of the chromatin or through the flanking sequences of the *MAT* locus. If the *Sc*Ho protein causes lethality by preventing repair at *MAT*, it is perhaps because it gets stuck at *MAT*, after performing the DSB, preventing recruitment of recombination proteins and thus repair of the locus. In *S. cerevisiae*, it is possible to follow the fate of repairable and non-repairable *Sc*Ho-cuts at *MAT* by Southern-blot analysis from a time-course experiment of the wild-type strains, overexpressing *Sc*Ho [37,38]. S3 Fig shows that, upon Ho induction, we are not able to visualize cut chromosomal arms at *MAT*. This could be explained if *Sc*Ho stays bound on the DNA end(s) and prevents HR at this locus: continuous resection will degrade DNA ends so that the probe cannot hybridize anymore. Indeed, in *S. cerevisiae*, unrepairable *Sc*Ho-cuts are more extensively degraded than repairable ones [37]. Nonetheless, we can note that we clearly observe the parental uncut band before induction and that this band disappears completely as soon as we start the experiment, thus confirming that the *MAT* locus is efficiently cut in our system.

Outside of its role in inducing a lethal DSB at *MAT*, *Sc*Ho displays further toxicity in *C. glabrata*. Strains that are inconvertible for the three *MTL*s exhibit a survival of 61% at the most, and lethality is strongly reduced by the deletion of one *MTL*, whatever its position, *HML*, *MAT* or *HMR*. The difference between those two types of strains is the number of inconvertible (*Inc*) Ho sites present at the *MTL* loci. One explanation for this would be that the *Sc*Ho protein binds the *Inc* Ho sites and gets stuck there, in a way that is toxic for *C. glabrata* cells. *Sc*Ho probably has a high affinity for the *C. glabrata* Ho sites since we know that they are cut very efficiently and rapidly. In the case of an *Inc* site, the protein may be stuck there because the substrate is not transformed into a product. Indeed, we hypothesize that the cut releases the endonuclease and as we speculated above, this is prevented at the *MAT* locus, even after cutting, possibly by the chromatin structure. On the *Inc* sites, the binding of *Sc*Ho does not induce massive cell death but could, for example, physically hinder replication forks and thus disrupt DNA replication and cell division. Performing a ChIP-PCR on the Ho protein to examine its binding on the three *MTL* loci would allow us to better explore this aspect.

To put our results back into an evolutionary perspective and explore the link between switching and sexuality, we can turn to the other *Nakaseomyces* species. This group comprises *C. glabrata*, two other pathogens, *Candida nivariensis* and *Candida bracarensis* and three environmental species, *N. delphensis*, *Candida castelli*, *Nakaseomyces bacillisporus*. This clade contains species both varied in their lifestyle and in their lifecycle; all *Candida* species are described as asexual and haploid and in the two sexual species, *N. delphensis* is an obligate haploid while *N. bacillisporus* is an diploid [27]. *N. delphensis* undergoes natural Ho-induced mating-type switching, just like *S. cerevisiae* [35]. It is striking that none of the asexual species exhibit switching, thereby reinforcing the notion that switching is a mechanism that favours sexual reproduction. It is also remarkable, in such a case, that all asexual species have conserved the three *MTL* and a highly similar *HO* gene [25,36,39] while switching is supposed to be a mechanism that favours sexual reproduction. It is understandable that such a dangerous mechanism, involving a chromosomal DSB, would be lost if it is not essential. It has been hypothesized that *C. glabrata* sometimes undergoes switching [40,41]. These events could indeed be the result of very rare Ho-induced cuts or could also be fortuitous gene conversion events independent of the Ho protein (replication accident, repair of accidental DSBs etc). If we accept that this system is largely non-functional in *C. glabrata* and possibly other asexual species, the question of why they have kept both *HO* and the *MTL* remains open. In addition, it has been reported that the *HO* gene is

under purifying selection in the population of *C. glabrata* strains analysed [25]. This may suggest that both Ho and the *MTLs* have acquired another function in the asexual *Nakaseomyces*. In *S. cerevisiae*, the *MTL* loci are one of the very few structured regions of the genome [42] and there remains an intriguing possibility that the conserved function of these *MTLs* and the Ho protein would be in structuring the chromosomes bearing them. Further studies of the 3D structure of the *C. glabrata* nucleus would shed light on this point.

## Materials and methods

### Strains, cultures, and transformation

*C. glabrata* strains used in this study are listed in Table 1. Strains are grown in broth or on plates at 28°C in YDP (non-selective, 1% Yeast Extract, 1% Peptone, 2% glucose), in Synthetic Complete medium lacking uracil (SC-Ura, 0.34% Yeast Nitrogen Base without amino acids, 0.7% ammonium sulfate, 2% glucose, supplemented with adenine and all amino acids except uracil) or in Synthetic Complete medium lacking uracil, methionine, and cysteine (induction conditions for the *MET3* promoter, SC-Ind, 0.34% Yeast Nitrogen Base without amino acids, 0.7% ammonium sulfate, 2% glucose, supplemented with adenine and all amino acids except methionine and cysteine). For selection of transformants of the Ho plasmid or Cas9 plasmid and maintenance in repressive conditions for the *MET3* promoter, strains are grown in SC-Ind supplemented with 2 mM each of methionine and cysteine (SC-rep) and in YPD supplemented with 2 mM each of methionine and cysteine (YDP-Rep) when repression but no selection is needed. For SC-Rep, medium is buffered by 10 mL of $Na_2HPO_4$ 0.05 M and $NaH_2PO_4$ 0.95 M per liter. For *URA3* marker counter-selection, yeast strains are grown on 5-FOA medium (SC-Ura supplemented with 1 g/L of 5-fluoroorotic acid (5-FOA) and 50 mg/L of uracil).

Transformation is done according to the "one-step" lithium acetate transformation protocol from [43].

### Induction of mating-type switching by *Sc*Ho

The *HO* gene from *S. cerevisiae* is cloned into the pCU-MET3 plasmid under the *MET3* promoter (p7.1, S2 Table) [44] and protocol for solid induction is detailed in [26]. For time-course of induction in liquid medium, transformants are grown overnight in liquid SC-Rep medium, counted, washed and resuspended in sterile water at $4.10^7$ cells/mL. 100 μL is used to inoculate 40 mL of liquid SC-Ind medium and the culture is placed at 28°C with agitation. For each time point, a sample of the culture is counted under the microscope, diluted and plated on SC-Rep plates. Each strain that was analysed in a time-course, were also transformed with the plasmid that does not contain *ScHO*, pYR32 plasmid, cells were diluted, counted and plated, allowing the normalization of the survival in Ho induction.

### Induction of mating-type switching by CRISPR-Cas9

We used the inducible CRISPR-Cas9 system for *C. glabrata* from [31] using plasmid pCGLM1. We cloned into pCGLM1 a sequence corresponding to a guide RNA (gRNA) targeting the Ya sequence (S1 Table), giving rise to plasmids pCGLM1-Ya2.

Induction of Cas9-DSB was then performed as inductions of the *ScHO* gene done with p7.1 (see above).

### Cell viability estimation

Different dilutions of cultures, containing between 200 to $10^6$ cells, are spread on both inductive and repressive media. When the survival rate is over 20%, cell viability is determined

directly as the ratio of the number of colonies counted on inductive medium to the number of colonies counted on repressive medium, for the same dilution. When the survival rate is under 1%, colonies are confluent on repressive medium at the same dilution where several colonies can be observed on induction medium. Thus, survival rate is measured by first comparing the number of colony-forming units (CFU) on inductive medium with the theoretical number of cells plated, as estimated by counting on a Thoma counting chamber. This is then corrected by the ratio of CFU to the number of cells counted, estimated by plating 200 cells on repressive medium. All the values were obtained from at least four independent transformants. Colonies number from a minimum of 2 to a maximum 746 was counted on plates. Numerical data used for drawing graphs is shown in S3 Table.

## Determining the genotype at *MTL* loci

The genotype at each *MTL* locus is determined by PCR, when needed, directly on colonies [26] using specific primers: the forward primer is located upstream of the locus (ensuring specificity of the locus screened; *HML*, *MAT* or *HMR*) and a reverse primer located precisely on the Ho site (ensuring specificity of the information carried by the locus; alpha or a and wt or inc) (S1 Table, S2 Fig). In most induction experiments, we did not check switching at *Inc* loci since preliminary experiments showed that there was no switching, indicating that the alpha-inc and a-inc Ho sites are not cut after Ho induction. This genotyping is performed on surviving colonies directly on induction plates (for solid induction) and on repression plates (for liquid induction). As previously shown, PCRs often reveal that most surviving colonies are mixed for genotypes at *MTL*s [26]. In the case of Ho inductions, we have already shown that sub-cloning of such mixed colonies yields more than 80% of switched pure clones [26]. Therefore, sub-cloning has not been done on any Ho induction in this work. In the case of Cas9 induction, since we had never used this system for switching *MTL*s, and we have decided to sub-clone mixed colonies in order to assess the true efficiency of mating-type switching.

## Construction of strains

We mutated the Ho sites in the region known to be essential for Ho cutting in *S. cerevisiae* [30], as shown on S1 Fig, yielding loci *HML-inc*, *MAT-inc* and *HMR-inc*. Modification of *HML*, *MAT*, or *HMR* loci was realized either by marker selection (pop-in/pop-out) [45] or by mating-type switching upon *HO* gene expression or by use of CRISPR-Cas9. The three methods are detailed below. Primers and plasmids are listed in S1 and S2 Tables, respectively. Method used to construct each strain is listed in S4 Table.

**Construction of PCR fragments and plasmids for pop-in.** In order to integrate the *URA3* marker at the targeted locus (pop-in), we amplified the *URA3* gene from *S. cerevisiae* under its own promoter by PCR using primers Sc-URA3-F and Sc-URA3-R and, YEp352 as template. The PCR fragment was cloned into the *EcoR*V-digested pBlueScript, giving rise to pURA (S2 Table).

To direct integration of the *URA3* marker at the targeted locus, here the *MTL* loci *HML*, *MAT* or *HMR*, the 5' and 3' flanking regions was added to the *URA3* marker in multiple steps.

First, the Z sequence, shared by the three *MTL* loci, was amplified by PCR using primers 68/70 and HM100 strain DNA as template (S1 Table). Primers 68 and 70 contain *BamH*I and *EcoR*I restriction sites, respectively, to allow cloning of the Z PCR fragment upstream of the *URA3* marker into pURA, giving rise to pZU (S2 Table).

Second, Ya and Yalpha sequences were amplified on strain HM100 by PCR, using primers 73/72 and 74/69 respectively (S1 Table). Primers 73 and 72 contain *Hind*III and *Sal*I restriction sites, respectively, in order to clone the Ya PCR fragment downstream of the *URA3* marker

into pZU, giving rise to pZUA (S2 Table). The *Sal*I restriction site was added to primer 69 and no restriction site was added to primer 74 as the Yalpha PCR fragment already contains the *Hind*III restriction site 38 bp from the 5' of the fragment. Thus, the Yalpha PCR fragment, digested by both *Sal*I and *Hind*III, was cloned downstream of the *URA3* marker into pZU to give rise to pZUAlpha (S2 Table).

Amplification by PCR, using universal primers M13F/M13R, on both pZUA and pZUAlpha plasmids, led to ZUA and ZUAlpha fragments, respectively. These fragments have been used for targeting *HML*, MAT or *HMR* loci (S2 Table) and Ura+ transformants were selected on SC-Ura. Correct integration of the fragment was checked by PCR.

**Construction of plasmids and PCR fragments for pop-out.** The *URA3* marker is removed (pop-out) from the target locus by homologous recombination with a DNA fragment derived from the upstream and downstream sequences of that locus (S4 Table).

In order to replace the wild-type Ho site in the different *MTL* loci, by the inconvertible-mutated Ho site, we constructed two plasmids; pZA-inc and pZalpha-inc (S2 Table). The pZA-inc plasmid (without *URA3* gene) results from double digestion of pZUA by *EcoR*I and *Hind*III and ligation after Klenow fill-in. The pZAlpha-inc plasmid (without the *URA3* gene) was constructed by cloning the *BamH*I/*EcoR*I-digested Z fragment and the *EcoR*I/*Sal*I-digested Yalpha fragment into the pBlueScript double digested by *BamH*I and *Sal*I. Amplification by PCR using primers M13F/M13R, from both pZA-inc and pZAlpha-inc plasmids, lead to the ZA-inc and ZAlpha-inc fragments that have been used for pop-out. The comparison of wild-type and inconvertible Ho sites is presented in S1 Fig.

In addition to the construction of *Inc* sites, we have also used strains with deletion of *MTLs* (Table 1). Strains with deletions of *MAT* and/or *HML* were directly obtained from [22] (Table 1) and Inc sites were introduced in those strains when needed (S4 Table). In addition, we constructed deletion of *HMR* in strains BG87 and CGM390. After pop-in of *URA3* at *HMR*, amplification of upstream and downstream sequences (500 bp each) of *HMR* was performed on strain BG87, using primers Up-HMR-F/Up-HMR-R and Down-HMR-F/ Down-HMR-R, respectively (S1 Table). Primer Up-HMR-R contains 40 bp of homology to the 5' end of the downstream PCR fragment. These two fragments were then combined by fusion PCR using primers Up-HMR-F and Down-HMR-R, giving rise to the *Δhmr* fragment (S4 Table).

As shown in S4 Table, other fragments for pop-out experiments were obtained by direct PCR on genomic DNA.

About 1 μg of each pop-out fragment was used to transform Ura+ strains, which were then plated onto YPD, grown for 24 hrs and replica-plated onto 5-FOA plates. Resulting 5-FOA$^R$ colonies were checked by PCR for correct removal of the *URA3* marker, and the locus sequenced.

**Strains obtained by mating-type switching.** When possible, we took advantage of the efficient mating-type switching induced by *ScHO* to transpose the inc-Ho site mutation from one *MTL* to another, instead of doing pop-in/pop-out transformations as above. For example, an *HMLalpha-inc* locus can easily be used as template, during gene conversion, to repair either *MAT* wt or *HMR* wt. In addition, extra-chromosomal copies of either *MATa-inc* or *MATalpha-inc* were also used as templates for mating-type switching of *MTL* loci, in order to insert inc-*Ho* sites. These copies were introduced in the p7.1 plasmid, as follows. Plasmid p7.1 [26] was digested by K*pn*I, and *MATa-inc* and *MATalpha-inc* sequences were amplified by PCR using primers Up-Rec-MAT-F/Down-Rec-MAT-R on strains YL09 and YL07, respectively (Table 1 and S1 Table). Both primers share, respectively, 40 bp of homology to the ends of the K*pn*I-digested plasmid. This allows PCR fragment cloning in p7.1, at the K*pn*I restriction site, by homologous recombination in *E. coli* [46]. Correct assembly was confirmed by both analytic colony PCR and restriction digests.

Expression of Ho is induced in strains that are targeted for modification, either from the p7.1 plasmid, when a genomic *MTL* locus is used as template, or from p7.1-derived plasmids that contain a copy of *MATa-inc* or *MATalpha-inc*. Final loci are checked by PCR and sequencing.

## Southern-blot analysis

Genomic DNAs were prepared using the Qiagen genomic DNA kit, according to manufacturer's instructions. 2 µg DNA was subjected to enzymatic digestions and protocol for Southern-blot is detailed in [26]. Primers used for probe PCR amplifications are given in S1 Table.

## Construction of the Δrad51 mutant using CRISPR-Cas9

The *Δrad51* mutant of strain HM100 was constructed with the CRISPR-Cas9 system on plasmid pJH-2972 (kind donation from J. Haber, https://protocolexchange.researchsquare.com/article/nprot-5791/v1). We cloned a sequence corresponding to a gRNA targeting the *RAD51* gene into plasmid pJH-2972 (S1 Table), giving rise to plasmid pJH-RAD51.

We amplified upstream and downstream sequences (500 bp each) of the *RAD51* CDS (*CAGL0I05544g*) on strain HM100 by PCR using primers Up-Rad51-F/Up-Rad51-R and Down-Rad51-F/Down-Rad51-R, respectively (S1 Table). Primer Up-Rad51-R contains 40 bp of homology to the 5' end of the downstream PCR fragment. These two fragments are then combined by fusion PCR using primers Up-Rad51-F and Down-Rad51-R, giving rise to the *Δrad51* fragment.

The strain was then co-transformed with both 1 µg of pJH-RAD51 and 1 µg of *Δrad51* fragment. Ura+ transformants were then selected on SC-Ura and checked for deletion at the *RAD51* locus by PCR. Deletion was confirmed by Southern blot analysis (S4 Fig) and by sequencing.

## Supporting information

**S1 Fig. Comparison of wild-type and mutated Ho sites of loci carrying Yalpha (A) or Ya information (B).** The wild-type Ho site is shown on top in blue letters, the mutated Ho site is shown below with mutated bp in red and deleted bp as dashes. Arrows indicate the Ho cleavage site.
(TIF)

**S2 Fig. Mating-type screened by PCR at *MAT* in different strains.** All strains are analyzed with primer pairs that are specific to *MATa*, *MATalpha*, *MATa-inc* and *MATalpha-inc*, respectively GS01/123, GS01/121, GS01/122 and GS01/120. Top left panel: amplification obtained on BG87 (*MATa*); bottom left panel: amplification obtained on YL05 (*MATa-inc*); top right panel: amplification obtained on HM100 (*MATalpha*); bottom right panel: amplification obtained on YL07 (*MATalpha-inc*). MM: Molecular Marker, GeneRuler 1 kb (Thermo Fisher Scientific Inc).
(TIF)

**S3 Fig. Southern-blot analysis of the DSB at the *MAT* locus in the wild-type HM100 strain.** Left panel: chemiluminescence image of blot of *Pst*I/*Xho*I digestion hybridized with a probe corresponding to 484 bp located 262 bp away from the first nucleotide of the Ho site. Diagram of probe in top middle panel. The regions are represented with restriction sites and size of expected fragment. Right panel: chemiluminescence image of blot of *Bgl*II/*EcoR*I digestion hybridized with a probe corresponding to 1,013 bp located 2,119 bp away from the first

nucleotide of the Ho site. Diagram of probe is in lower middle panel.
(TIF)

**S4 Fig. Molecular characterization of *Δrad51* mutations in HM100 by Southern blot hybridization.** Left panel: chemiluminescence image of blot of *Hind*III digestion. Right panel: chemiluminescence image of blot of *Nde*I/*Eco*RI digestion. The probe used is 1 kb long and is composed to the 500 bp upstream of the *RAD51* ORF fused to the 500 bp downstream of the *RAD51* ORF.
(TIF)

**S1 Table. Primers used in this work.** Fw: Forward; Rv: Reverse. The lowercase letters represent sequences with no homology to template DNA and reverse complete regions are indicated in uppercase.
(DOCX)

**S2 Table. Plasmids used in this work.**
(DOCX)

**S3 Table. Numerical data for figures.** Tables show percentage of survival calculated for each individual transformant in experiments represented as graphs. ND: Not done. SEM: Standard error of the mean. For Figs 4 and 5C number of colonies counted during time-course experiments is also shown.
(XLSX)

**S4 Table. Methods used for strain construction.**
(DOCX)

## Acknowledgments

We thank I. Castaño for providing strains CGM460, CGM390 and GM498. We thank members of our lab and the iGénolevures network ((IRN from the CNRS N˚0814) for stimulating discussions, Gilles Fischer, Fabienne Malagnac, Pierre Grognet, and anonymous reviewers for critical reading. We are also grateful to Pierre Grognet for his technical help with the Southern-blot.

## Author Contributions

**Conceptualization:** Laetitia Maroc, Youfang Zhou-Li, Stéphanie Boisnard, Cécile Fairhead.

**Formal analysis:** Laetitia Maroc.

**Investigation:** Laetitia Maroc, Youfang Zhou-Li, Stéphanie Boisnard.

**Methodology:** Youfang Zhou-Li.

**Supervision:** Cécile Fairhead.

**Writing – original draft:** Laetitia Maroc, Cécile Fairhead.

**Writing – review & editing:** Laetitia Maroc, Cécile Fairhead.

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
