## [Decision Letter · Decision Letter 0]

9 Mar 2020

Dear Cecile,

Thank you very much for submitting your Research Article entitled 'A single Ho-induced double-strand break at the MAT locus is lethal in Candida glabrata' to PLOS Genetics. Your manuscript was fully evaluated at the editorial level and by independent peer reviewers who are all experts in the field. The reviewers appreciated the attention to an important problem, but raised some substantial concerns about the current manuscript. Based on the reviews, we will not be able to accept this version of the manuscript, but we would be willing to review again a much-revised version. We cannot, of course, promise publication at that time.  The reviewers have provided ideas about how to restructure the manuscript, how to present the models or hypotheses being tested, and suggested additional experiments.  Given the extent of their reviews, this was on the edge of major revision/rejection.  I am sending to you the decision of major revision to provide an opportunity to address the reviews.  I appreciate that this may take some time, and thus am extending the time available from 60 to 120 days.  A resubmitted manuscript would be reviewed again by the same set of reviewers if available.

If you decide to revise the manuscript for further consideration at PLOS Genetics, please aim to resubmit within the next 120 days, unless it will take extra time to address the concerns of the reviewers, in which case we would appreciate an expected resubmission date by email to plosgenetics@plos.org.

[LINK]

We are sorry that we cannot be more positive about your manuscript at this stage. Please do not hesitate to contact us if you have any concerns or questions.

Yours sincerely,

Joseph Heitman, MD, PhD

Associate Editor

PLOS Genetics

Gregory P. Copenhaver

Editor-in-Chief

PLOS Genetics

Reviewer's Responses to Questions

**Comments to the Authors:**

Reviewer #1: This is a careful analysis of the fate of Candida glabrata cells when the Ho endonuclease is expressed in Candida. An earlier publication showed that expression of Ho generated DSBs that were substrates for recombination and mating type switching but that resulted in very drastic lethality. The result shown here by analyzing switching and lethality in mutant strains where Ho sites have been removed from MAT, HML or HMR is that cleavage at MAT alone is sufficient to generate lethality independent of the mating type information at MAT. The data are consistent with the DSB being poorly repaired when generated by Ho, since a DSB generated by CRISPR/CAS show low lethality.

The authors nicely show that lethality depends on having a Ho-sensitive MAT allele. Interestingly, the survival does not increase when both HMR and HML are Inc. This suggests that cleavage at these loci is not an important contributor to lethality but also that repair from these loci and then persistent re-cutting is probably not the mechanism of lethality since repairing with an INC substrate would be expected to increase survivability. Rather the data suggest that the cut at MAT is not easily repairable. Cutting at MAT by CRISPR/CAS results in efficient switching but no lethality. In colonies that result, 80-100% of cells have switched and are viable. This suggests that the DSB generated by Ho is specifically lethal.

These data are excellent and the conclusions are compelling and the genetic analysis with the engineered strains is elegant and informative, and the results are surprising. Some additional controls are needed, or if they exist should be shown, to see whether cleavage by Ho is occurring at HMR and HML. If these are actively being cut and gene converted by the other HM locus or by MAT, then the question is why there is no lethality. Are the survivors in YL04 YL05 and YL07 able to repair a DSB, or is the information between HML and HMR non-essential and simply lost, or are these loci not being cleaved at appreciable rates? The manuscript does not present quantitative data to show whether there is cleavage at HML and HMR and whether DSBs there are repaired by homologous recombination. This would help distinguish models for why MAT DSBs are so toxic.

The paper is a little anticlimactic since there is no final conclusion as to why the Ho generated DSB can't be repaired. it would be easily improved by direct assays demonstrating persistence of the DSB. If it is happening in 99% of cells, can the DSB be visualized by Southern, or quantitated by a PCR assay with a primer ligated to the break? If so, comparing DSB persistence at MAT HML and HMR would add significantly to the paper. Can a tagged version of Ho be used for an IP to show presence at the MAT locus and/or the HML and HMR loci?

Reviewer #2: This is an intriguing and well written manuscript that aims to understand the mechanism by which the Saccharomyces cerevisiae (Sc) HO endonuclease causes severe lethality when expressed in Candida glabrata (Cg) cells. It has implications for our understanding of how HO works, and the manuscript provides some interesting indications that HO has a function that goes beyond being a simple endonuclease.

In previous work (Boisnard et al 2015, ref. 22) the authors’ group discovered that expressing Sc HO in C. glabrata causes a high rate of cell death (99.9% of cells die). In that study, they proposed that the lethality was caused by simultaneous cutting of the Cg HMR and HML loci as well as the MAT locus, leading to absence of intact templates for DNA repair. However, in the current manuscript Maroc et al test this hypothesis and reject it. In a careful series of experiments in which they mutated the HO cleavage sites in Cg MAT, HML and/or HMR, they show that HO-cleavage of the MAT locus alone is sufficient to cause lethality, even though most of the few cells that survive HO expression have switched mating type. Maroc et al also show that, in contrast, cleavage of the Cg MAT locus by CRISPR-Cas9 does not cause lethality and leads to mating-type switching. So the lethality does not stem from the existence of a DSB per se, but from the existence of a DSB in association with HO, which is intriguing. Although the manuscript does not succeed in its goal of identifying the molecular cause of the incompatibility between Sc HO and Cg MAT, it goes a long way towards narrowing down the range of possible causes.

Major comment

L368-378 (end of Discussion section): The manuscript has made a substantial step towards understanding the toxic role of S. cerevisiae HO in C. glabrata, but we are left with the possibility that HO might be toxic because it attacks some unknown off-target (non MAT) site in the C. glabrata genome (L372-373). A fairly simple way to address this question experimentally would be to see if the lethality can be moved into S. cerevisiae, by moving the Cg MAT cleavage site into S. cerevisiae. Construct a haploid S. cerevisiae strain that is uncleavable at the Sc MAT locus (MAT-inc or similar), but which also carries the Cg cleavage site (~100 bp spanning the Cg Y/Z junction) somewhere in its genome, and then introduce HO from p7.1 to see if it causes lethality.

Minor comments

L24, L28, The taxonomic group can be called either Ascomycota (phylum) or Saccharomycotina (subphylum), but there is no ‘Ascomycotina’.

L40, add “…, whereas one DSB at HML or HMR is not.”

L44, change to “…, but that repair is prevented by S. cerevisiae’s Ho”.

L86, change to “… mat1, called mat2 and mat3”.

L88, also cite Rajaei et al, PMID 25313032.

L100, change “implies” to “involves”

L114, L221, L305: When referring to the RE, the manuscript cites reference 15 (a review article by Herskowitz 1988) several times. Surely there are more relevant references, such as PMID 8861911, 9335581.

L132-133: change to “a series of inconvertible (inc) C. glabrata strains … DNA breaks during induction of S. cerevisiae Ho … “

L152: I was surprised by the reference to data “not shown” here, because the title of this section is “HMR is cut by Ho …”. The authors do show data in support of HMR cleavage later in the manuscript (SL-CG9 results in Table 2) so it would be preferable to rearrange the order of the sections in Results / or merge some of them / or rephase L152 to say that evidence for cleavage of HMR will be presented in a subsequent section of the paper.

L232: This paragraph should not begin with “Similarly...” but “In contrast,… ” because the frequency of switching in the correct direction changes dramatically from 97% (MATa to MATalpha) to <14% (MATalpha to MATa).

L300: change to “cut by the S. cerevisiae Ho endonuclease”

L311-315: I agree with the authors that the mating-type switching system in C. glabrata is less active/efficient than in S. cerevisiae, and that the word “degenerated” is probably an accurate word to use. However Cg HO is not a pseudogene, which is what we would expect if it truly has no function. One could speculate that HO might have a second function in addition to being the mating-type switching endonuclease, and this second function causes the HO gene to be maintained in C. glabrata. In the population genomics data of Carrete et al (2018), did all the studied C. glabrata stains have an intact HO ORF, and was the nonsynonymous/synonymous substitution ratio for this gene <1, indicating purifying selection ?

L350: Change to “suggest that the S. cerevisiae Ho protein prevents DSB repair specifically at the MAT locus of C. glabrata”

L357-359. I didn’t understand the sentences “We know that mating-type borne by any of the MAT does not influence lethality as both HM100 and BG87 die at 99.99 % (Fig 1). Thus, only the location of the MAT locus could explain the specificity of lethality induced by Ho.” Do you mean “We know that lethality occurs in both MATa and MATalpha cells, because both HM100 and BG87 die at 99.99%”?

L361: ref 36 is the wrong reference.

L368: “Finally, we would like to discuss the toxic role of S. cerevisiae’s Ho, in C. glabrata, outside its role in lethal DSB at MAT.” Is this an hypothesis (you are proposing that it has an unidentified toxic role apart from DSB formation), or is it a statement of fact? If it is a statement of fact, what is the evidence that such a toxic role outside DSB formation exists?

The plasmid name p7.1 appears in the titles of several Figures and Tables, but it is not explained in the text until line 402. It should be introduced on approx. line 180.

Figure 1: Shouldn’t HML be colored red (alpha) in HM100 and HM100 Δrad51 ?

Figure 2: I found the X-axis title “Time of nuclease induction” confusing because it seems to indicate that the nuclease was induced at time T, where T = 0, 2, 4 hrs etc. But I think that you actually mean that the nuclease was induced at T=0 and then repressed at T = 2, 4, hrs etc. Change the title to “Duration of nuclease induction” ?

Figure 3B: The HO cut is shown as a blunt DSB, but it should be a 4-bp 3’ overhang. Also, the Y/Z junction is drawn in the wrong place: it should be between the TTTT and the CGCAACA on the bottom strand (if we permit a 1-bp mismatch in the Z region, CGCAACA in MATa and CGCAGCA in MATalpha).

Tables 2 and 3: For readers like me who are slow at mental arithmetic, it would be easier to relate the colony numbers in the Tables to the percentages in the text, if you showed percentages as well as numbers in the Tables. For example, for the SL-CG9 Cas9 experiment, Line 264 says that 87% of colonies presented MATalpha-inc and cites Table 3, but to find this information in Table 3 I have to calculate that 32/45 + 7/45 = 87%.

Reviewer #3: Mating type switch in Saccharomyces and related yeasts is a well-studied process, but many unanswered questions remain. The authors have addressed the puzzling retention of the mating-type switching apparatus in Candida glabrata, even though it is apparently asexual. Intriguingly, they showed that that they can induce switching in C. glabrata using CRISPR-Cas9, demonstrating that the all the components (apart from the endonuclease) are fully functional. However, when the S. cerevisiae HO endonuclease is overexpressed, cutting at MAT results in cell death. HML and HMR are also cut. These experiments are carefully performed, but the underlying biological mechanism effect is not clear.

1. I think that the CRISPR-Cas9 mediated induction of switching is one of the most interesting parts of this paper. It has been shown in S. cerevisiae, but this is the first time that it is clear that switching is fully functional in C. glabrata (apart from the endonuclease) and that cells can switch without inducing cell death. I suggest re-framing the paper, and starting from here.

2. RAD51 should be inactivated in the Cas9-activated strain, this would show more clearly if switching relies on homologous recombination.

3. The authors show that Cas9 induces switching but not cell death, whereas expression of ScHO induces switching and cell death. They suggest that Ho may remain at the cut ends. What happens if Cas9 and Ho are added/induced at the same time? This might help to work out the mechanism.

4. I think that the observation that expressing ScHO in K. delphensis does not induce cell death is important, and should be shown in the manuscript.

5. There seems to be an error in Fig. 1, the colors for the top two strains are not correct, surely they are HMLalpha?

Minor points

1. The start of the results section is hard to follow. It is assumed that the reader is very familiar with the earlier experiments. More details, for example how ScHO is expressed and how the target sites are changed, would be helpful. Switching at HMR should be shown, PLoS does not allow “data not shown”

2. There are some odd phrases. Line 138 might be better phrased as “ We find that even when HML and HMR are protected, introducing a DSB at MAT is sufficient…”

3. Line 124 has an extra “to”.

4. Line 128: unlike, rather than contrary

5. Line 85: Clarify what an “imprint” at mat1 means.

Reviewer #4: The manuscript ‘A single HO-induced double-strand break at the MAT locus is lethal in Candida glabrata’ seeks to follows up on an interesting result from a previous study that found lethality associated with heterologous S. cerevisiae HO expression in C. glabrata. The authors find that the rare cells that survive ScHO expression can indeed undergo mating type switching, although there is a preference for HML over HMR, regardless of the MAT genotype. Furthermore, mating type switching is dependent on Rad51 and homologous recombination. The authors construct a series of strains in which they disrupt the HO-cleavage sites at the MAT or cryptic mating type loci, HML and HMR and then measure cell lethality when cutting is blocked. Intriguingly, it is cleavage at the MAT locus, but not the cryptic mating type loci that results in a lethal phenotype. Furthermore, MAT-cleavage lethality is specific to HO cutting – the authors induce CAS9 cleavage at the MAT locus and find that lethality is abolished, but the frequency of mating-type switching is reduced compared to HO cleavage. While this is an interesting insight into the degradation of pseudo-homothallism in C. glabrata, the current presentation and organization of this manuscript makes it difficult to follow and does not provide enough context for non-expert readers, and the interpretation of the HO-specific lethality is tenuous, given the limitations of the experiment. Taken together, this work is not of sufficient interest to the broad range of PLoS Genetics readers in its current state. Furthermore, there are several experimental considerations that should be taken into account.

1) There are currently three main figures – but many aspects of figures 1 and 2 and not discussed until much later in the manuscript. As a reader this was very difficult to discern what was going on. I suggest rearranging the figures. The first part of figure 1 (all the strains that have intact cleave sites) should be presented with figure 2a. The second figure should be the constructed strains that cannot cut at the various loci and their resulting survival. The third figure should be regarding the Cas9-induced cleavage and include figure 2b (which is currently not discussed or referred to in the text).

2) The CAS9-induced cleavage is a very cool experiment, but given that the cleavage site is different, if only by an 18-bp shift, it is hard to directly compare HO- and CAS9- induced cleavage. If the authors designed additional gDNAs that could cut at many different places in the MAT locus this would bolster their argument that lethality is specific to HO. Alternatively, if the HO cleavage site was moved to different locations within the MAT locus and still resulted in lethality, this also would better support that HO-cleavage is distinct from other DSBs and not due to exact location of the cut.

3) The authors refer to cell viability as the survival rate, but my interpretation of the experiment is that this is a frequency of surviving cells relative to the total number in the population, and thus it would be more appropriate to refer to this as a frequency, or simply % survival.

4) I’m a little confused about the ‘mixed’ mating-type analysis – does this suggest that some surviving cells are able to survive because they are delayed in HO cleavage and thus, when HO cuts it can switch using HML or HMR within that colony? Also, the sub-cloning results seemed inconsistent with the original analysis. The original analysis showed that ~20% of colonies were mixed, but then the subcloning indicated that there were no MATa-inc isolated? Clarification regarding this experiment and analysis is needed.

5) I’m not exactly sure what audience this paper was written for. Certainly for an expert audience in yeast mating-type switching, but aside from that, it’s unclear if the scope of the paper is to investigate the molecular mechanism by which switching is lethal in C. glabrata or to investigate the evolutionary trajectory as to why C. glabrata is seemingly asexual. I think it is likely both, but these ideas are not woven together and developed in this manuscript. Most experiments in the results are not introduced with the specific question they are designed to address, which leaves the reader guessing as to what we are trying to learn. Providing more background and context for the motivation for this study (beyond just stating that an earlier study showed heterologous expression of ScHO is lethal in C. glabrata) is needed.

**Have all data underlying the figures and results presented in the manuscript been provided?**

Reviewer #1: Yes

Reviewer #2: Yes

Reviewer #3: No: Some experiments listed as "not shown"

Reviewer #4: No: NA. No large-scale experiments were discussed in this study. Also, a number of results were indicated as 'data not shown'

PLOS authors have the option to publish the peer review history of their article (what does this mean?). If published, this will include your full peer review and any attached files.

Reviewer #1: No

Reviewer #2: No

Reviewer #3: No

Reviewer #4: No

---

## [Decision Letter · Decision Letter 1]

17 Jul 2020

Dear Cecile,

Thank you very much for submitting your revised Research Article entitled 'A single Ho-induced double-strand break at the MAT locus is lethal in Candida glabrata' to PLOS Genetics. Your manuscript was fully evaluated at the editorial level and by the same set of independent peer reviewers who reviewed the original submission. The reviewers appreciated the attention to an important problem, and found the revised manuscript greatly improved.  This said, two of the four still found that additional experiments might be needed to address models, hypotheses advanced, and two others felt the manuscript would benefit from having some data removed and not presented.  Based on the reviews, we will not be able to accept this version of the manuscript, but we would be willing to review again a revised version. This would fall in between minor and major revision.  We appreciate that this isn't the response that authors are hoping to receive, and appreciate your continued interest in publishing these studies in PLOS Genetics.  We hope that you find the reviews and comments helpful and constructive, and look forward to receiving a revised manuscript that addresses the reviews.

If you decide to revise the manuscript for further consideration at PLOS Genetics, please aim to resubmit within the next 60 to 90 days, unless it will take extra time to address the concerns of the reviewers, in which case we would appreciate an expected resubmission date by email to plosgenetics@plos.org.

[LINK]

We are sorry that we cannot be more positive about your manuscript at this stage. Please do not hesitate to contact us if you have any concerns or questions.

Yours sincerely,

Joseph Heitman, MD, PhD

Associate Editor

PLOS Genetics

Gregory P. Copenhaver

Editor-in-Chief

PLOS Genetics

Reviewer's Responses to Questions

**Comments to the Authors:**

Reviewer #1: This revision appropriately addresses many reviewers' concerns. Demonstration of efficient cleavage and repair at HML and HMR adds significantly to the paper. It highlights the the key question of why HO generated breaks at MAT are lethal while cleavage at HML or HMR is non lethal as long as there is a template to repair from.

To limit the reasons for this locus specific lethality, an additional experiment would be helpful.

Replace the entire HML or HMR locus with the entire MAT locus, and determine if cleavage here results in lethality. The MAT/HML/HMR loci are not identical and perhaps sequence differences in the locus are responsible. In any case, it would be worth eliminating this possibility.

This proposed additional experiment is similar to the suggestion from a previous reviewer to test lethality of MAT +ScHO in S. cerevisiae, and eliminates a trivial explanation for the striking difference in lethality for HO-generated cleavage at MAT and HML/HMR

Reviewer #2: I am happy with the changes that have been made to the manuscript. The authors have addressed the point about residual lethality that I raised in my previous review.

The revised manuscript has been restructured and is presented in a clearer way than originally. The experiments and arguments are logical and show that:

ScHO endonuclease is lethal when expressed in C. glabrata.

ScHO cleaves all three Cg MTL loci (MAT, HML and HMR).

The lethality of ScHO is due specifically to it cutting the Cg MAT locus, no HML or HMR.

A CRISPR cut at Cg MAT is not lethal and is repaired by mating-type switching.

One small comment is that Figure 1 would be clearer if the HO sites were marked on the sequences, e.g. by underlining, because they are in different orientations for the two species. Also, please check the sequence shown for the Z1 region at Sc HMRa: it shows a 3-bp insertion compared to the Z1 sequences of Sc HMLalpha and MATalpha, and I think this is an error.

In Figure 5, I am not sure that inclusion of the experiments with additional CRISPR guide RNAs (Ya1, Ya3, Ya4) has added anything to the manuscript and I would suggest dropping these.

Reviewer #3: Many, though not all, of the reviewers' requests have been addressed. The most useful are the addition of data in Table 2 indicating that HML and HMR cutting is efficient, and the experiment in Fig. 6 addressing possible lethality caused by cutting at sites other than MAT.The introduction is also much improved. However, i don't think that adding extra gRNAs (Figure 5) has helped. Only one gRNA (the one described in the original submission) induced cutting. Looking at Fig. 5, this is not surprising. For Ya1 and Ya3, the cut site is some distance from the guide, and there is a lot of evidence that distance reduces efficiency. For Ya4, the PAM site is a very inefficient NAG, and this is part of the gRNA sequence. I don't think Cas9 will cut within a guide sequence. I understand that the authors were trying to address a reviewer's comment, but the added experiments add more confusion rather than clarifying. I suggest reverting to the original.

Reviewer #4: The revised manuscript ‘A single HO-induced double-strand break at the MAT locus is lethal in Candida glabrata’ has adequately addressed the most of the considerations I posed during its initial review. The added discussion regarding the evolution and mechanism of mating type switching is welcome, as is the inclusion of the experimental questions in the results section.

There are three outstanding concerns with the revised manuscript.

1) It is presumed that the C. glabrata HO endonuclease is functional, yet doesn’t result in lethality when expressed, but there is 58% identity (82% similarity) with the S. cerevisiae homolog. Has there been biochemical analysis of its cutting efficiency already reported? If not, an EMSA experiment, could demonstrate whether both endonucleases have the capacity to cut at the same efficiency.

2) The low lethality when CAS9 is expressed – could this also be due to low expression or inefficient cutting? Clearly it is capable of cutting, otherwise you wouldn’t see any switching, but how can one distinguish between low efficiency and strong selection for escaping the cleavage of MAT (such as by mutating the plasmid as the authors suggest)?

3) I still find presentation of the figures to be problematic with the text. For example, in Figure 2 there are two strain that are initially discussed (HM100 and HM100 �rad51; lines 178-182) and the other four strains that are not discussed until after Figure 3 has been presented (lines 222-232). Likewise, the text regarding data presented in Figure requires us to go back to Figure 3 to have a control to compare it. For experts in the field, this may not be an issue, but for those new to the field or still in their scientific training, this is very confusing and can be easily rectified. For example, in the latter case, adding the data from strain SL09 to Figure 6 with a note in the legend that it is the same data as in Figure 3 would suffice and would allow for the statistical significance to be conveyed on the graph in addition to in the text.

Minor notes:

Line 161 is written in the present tense, yet the rest of the section is written in the past tense.

Line 344: ‘inducting’ should be inducing?

Line 363: what is meant by ‘some sexual identity of cells in maintained in C. glabrata’?

**Have all data underlying the figures and results presented in the manuscript been provided?**

Reviewer #1: Yes

Reviewer #2: Yes

Reviewer #3: Yes

Reviewer #4: Yes

PLOS authors have the option to publish the peer review history of their article (what does this mean?). If published, this will include your full peer review and any attached files.

Reviewer #1: No

Reviewer #2: No

Reviewer #3: No

Reviewer #4: No

---

## [Editor Report · Decision Letter 2]

12 Sep 2020

Dear Cecile,

Congratulations!  We are pleased to inform you that your revised manuscript entitled "A single Ho-induced double-strand break at the MAT locus is lethal in Candida glabrata" has been editorially accepted for publication in PLOS Genetics.

Thank you again for supporting open-access publishing; we are looking forward to publishing your work in PLOS Genetics!  We appreciate your careful revision of the manuscript, including additional experimental analysis, and we hope that our review process has been a professional, collegial, and positive experience.  We very much appreciate your entrusting this outstanding manuscript to PLOS Genetics, and look forward to seeing this published.

Yours sincerely,

Joe

Joseph Heitman, MD, PhD

Associate Editor

PLOS Genetics

Gregory P. Copenhaver

Editor-in-Chief

PLOS Genetics

Comments from the reviewers (if applicable):

**Data Deposition**

http://datadryad.org/submit?journalID=pgenetics&manu=PGENETICS-D-20-00064R2

**Press Queries**

---

## [Editor Report · Acceptance letter]

8 Oct 2020

PGENETICS-D-20-00064R2 

A single Ho-induced double-strand break at the *MAT* locus is lethal in *Candida glabrata*

Dear Dr Fairhead, 

We are pleased to inform you that your manuscript entitled "A single Ho-induced double-strand break at the *MAT* locus is lethal in *Candida glabrata*" has been formally accepted for publication in PLOS Genetics! Your manuscript is now with our production department and you will be notified of the publication date in due course.

With kind regards,

Jason Norris

PLOS Genetics

On behalf of:
